# Forward Learning with Top-Down Feedback: Empirical and Analytical Characterization

**Ravi Srinivasan**[1,2]**, Francesca Mignacco**[3,4]**, Martino Sorbaro**[5,6]**,**
**Maria Refinetti**[7,8]**, Avi Cooper**[9]**, Gabriel Kreiman**[10,11,†]**, Giorgia Dellaferrera**[1,2,5,10,11,‡]

[1] IBM Research Europe, Zurich, [2] ETH Zurich, [3] Joseph Henry Laboratories of Physics, Princeton University, [4] Initiative for the Theoretical Sciences, Graduate Center, City University of New York,
[5] Institute of Neuroinformatics, University of Zurich and ETH Zurich, [6] AI Center, ETH Zurich,
[7] Laboratoire de Physique de l'Ecole Normale Supérieure, Université PSL, CNRS, Sorbonne Université, Université Paris-Diderot, Sorbonne Paris Cité, [8] IdePHICS laboratory, École Fédérale Polytechnique de Lausanne (EPFL), [9] Massachusetts Institute of Technology, [10] Center for Brains, Minds and Machines,
[11] Children's Hospital, Harvard Medical School
[†] gabriel.kreiman@tch.harvard.edu
[‡] giorgia.dellaferrera@gmail.com

## Abstract

"Forward-only" algorithms, which train neural networks while avoiding a backward pass, have recently gained attention as a way of solving the biologically unrealistic aspects of backpropagation. Here, we first address compelling challenges related to the "forward-only" rules, which include reducing the performance gap with backpropagation and providing an analytical understanding of their dynamics. To this end, we show that the forward-only algorithm with top-down feedback is well-approximated by an "adaptive-feedback-alignment" algorithm, and we analytically track its performance during learning in a prototype high-dimensional setting. Then, we compare different versions of forward-only algorithms, focusing on the Forward-Forward and PEPITA frameworks, and we show that they share the same learning principles. Overall, our work unveils the connections between three key neuro-inspired learning rules, providing a link between "forward-only" algorithms, *i.e.,* Forward-Forward and PEPITA, and an approximation of backpropagation, *i.e.,* Feedback Alignment.

## 1 Introduction

In machine learning (ML), the credit assignment (CA) problem refers to estimating how much each parameter of a neural network has contributed to the network's output and how the parameter should be adjusted to decrease the network's error. The most commonly used solution to CA is the Backpropagation algorithm (BP) (Rumelhart et al., 1995), which computes the update of each parameter as a derivative of the loss function. While this strategy is effective in training networks on complex tasks, it is problematic in at least two aspects. First, BP is not compatible with the known mechanisms of learning in the brain (Lillicrap et al., 2020). While the lack of biological plausibility does not necessarily represent an issue for ML, it may eventually help us understand how to address shortcomings of ML, such as the lack of continual learning and robustness, or offer insights into how learning operates in biological neural systems (Richards et al., 2019). Second, the backward pass of BP is a challenge for on-hardware implementations with limited resources, due to high memory and power requirements (Khacef et al., 2022; Kendall et al., 2020).

These reasons motivated the development of alternative solutions to CA, relying on learning dynamics that are more biologically realistic than BP. Among these, several algorithms modified the feedback path carrying the information on the error (Lillicrap et al., 2016a; Nøkland, 2016; Akrout et al., 2019; Clark et al., 2021) or the target (Lee et al., 2015; Frenkel et al., 2021) to each node, while maintaining the alternation between a forward and a backward pass. More recently, "forward-only" algorithms were developed, which consist of a family of training schemes that replace the backward pass with a second forward pass. These approaches include the *Forward-Forward* algorithm (FF)

(Hinton, 2022) and the *Present the Error to Perturb the Input To modulate the Activity* learning rule (PEPITA) (Dellaferrera & Kreiman, 2022). Both algorithms present a clean input sample in the first forward pass (*positive phase* for FF, *standard pass* for PEPITA). In FF, the second forward pass consists in presenting the network with a corrupted data sample obtained by merging different samples with masks (*negative phase*). In PEPITA, instead, in the second forward pass the input is modulated through information about the error of the first forward pass (*modulated pass*). For simplicity, we will denote the first and second forward pass as *clean* and *modulated* pass, respectively, for both FF and PEPITA. These algorithms avoid the issues of weight transport, non-locality, freezing of activity, and, partially, the update locking problem (Lillicrap et al., 2020). Furthermore, as they do not require precise knowledge of the gradients, nor any non-local information, they are well-suited for implementation in neuromorphic hardware. Indeed, the forward path can be treated as a black box during learning and it is sufficient to measure forward activations to compute the weight update.

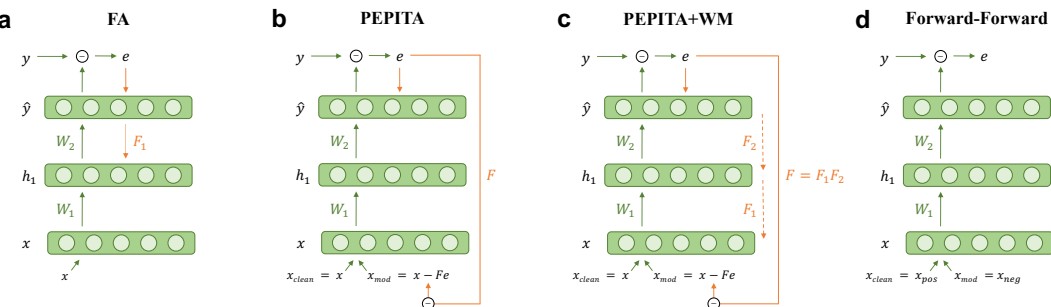

Figure 1: Different error transportations and WM configurations. Green arrows mark forward paths and orange arrows indicate error paths. **(a)** Feedback alignment (FA). **(b)** Present the Error to Perturb the Input To modulate Activity (PEPITA). **(c)** PEPITA with WM. **(d)** Forward-Forward (FF).

Despite the promising results, the "forward-only" algorithms are still in their infancy. First, neither Hinton (2022) nor Dellaferrera & Kreiman (2022) present an analytical argument explaining why the algorithms work effectively. Second, they present some biologically unrealistic aspects. For instance, the FF requires the generation of corrupted data and their presentation in alternation with clean data, while PEPITA needs to retain information from the first forward pass until the second forward pass, therefore the algorithm is local in space but not local in time. Third, the algorithms present a performance gap when compared with other biologically inspired learning rules, such as feedback alignment (FA). Our work addresses the mentioned limitations of the "forward-only" algorithms, focusing on PEPITA, and compares the FF and the PEPITA frameworks. Here, we make the following contributions: *(i)* We show that PEPITA effectively implements "feedback-alignment" with an adaptive feedback (AF) matrix that depends on the upstream weights (Sec. 3.1); *(ii)* Focusing on PEPITA, we use the above result to analytically characterize the online learning dynamics of "forward-only" algorithms and explain the phenomenon of alignment of forward weights and top-down connections observed in original work (Sec. 3.2); *(iii)* We demonstrate that PEPITA can be applied to networks deeper than those tested by Dellaferrera & Kreiman (2022) (Sec. 4); *(iv)* We propose two strategies to extend the weight mirroring (WM) method to training schemes other than FA and show that PEPITA enhanced with WM achieves better alignment, accuracy, and convergence (Sec. 4); *(v)* We demonstrate that PEPITA can be approximated to a rule containing a sum of Hebbian and anti-Hebbian terms, allowing space- and time-locality (Sec. 5.1 and Appendix B); *(vi)* We analytically compare the Hebbian approximation of PEPITA with the Forward-Forward algorithm (Hinton, 2022) with top-down feedback (Sec. 5.2).

## 2 BACKGROUND AND RELEVANT WORK

### 2.1 BIO-INSPIRED LEARNING ALGORITHMS

Several training rules for artificial neural networks (ANNs) have been proposed to break the weight symmetry constraint required by BP (Lillicrap et al., 2016a; Nøkland, 2016; Liao et al., 2016; Nøkland & Eidnes, 2019; Frenkel et al., 2021; Hazan et al., 2018; Kohan et al., 2018; 2022; Meulemans et al., 2021; Halvagal & Zenke, 2023; Journé et al., 2022; Launay et al., 2020). Furthermore,

Table 1: Biological properties and test accuracy [%] on MNIST achieved by a non-exhaustive selection of bio-inspired alternatives to BP. We report the highest accuracy documented in the referenced papers. Legend: ✗ = aspect not solved; ✓ = aspect fully solved; ∼ = aspect partially solved.

| LEARNING RULE | REF. | FORWARD ONLY | WEIGHT TRANSPORT FREE | LOCAL | ACTIVITY FREEZING | UPDATE UNLOCKED | MNIST |
|---|---|---|---|---|---|---|---|
| FA | LILLICRAP ET AL. (2016B) | ✗ | ✓ | ✗ | ✗ | ✗ | 98.74 |
| DFA | NØKLAND (2016) | ✗ | ✓ | ✗ | ✗ | ∼ | 98.55 |
| DRTP | FRENKEL ET AL. (2021) | ✗ | ✓ | ✓ | ✗ | ✓ | 98.52 |
| DTP-$\sigma$ | ORORBIA & MALI (2019) | ✗ | ✓ | ✓ | ✗ | ✓ | 97.62 |
| LRA | ORORBIA & MALI (2019) | ✗ | ✓ | ✓ | ∼ | ✓ | 98.03 |
| RLRA | ORORBIA ET AL. (2020) | ✗ | ✓ | ✓ | ✓ | ✓ | 98.18 |
| EFP | KOHAN ET AL. (2018) | ✓ | ✓ | ✓ | ✓ | ✗ | 98.24 |
| PEPITA | DELLAFERRERA & KREIMAN (2022) | ✓ | ✓ | ✓ | ✓ | ∼ | 98.29 |
| FF | HINTON (2022) | ✓ | ✓ | ✓ | ✓ | ✓ | 98.84 |
| PFF | ORORBIA & MALI (2023B) | ✓ | ✓ | ✓ | ✓ | ✓ | 98.66 |
| SP | KOHAN ET AL. (2022) | ✓ | ✓ | ✓ | ✓ | ✓ | 98.62 |
| PEPITA+WM | OURS | ✓ | ✓ | ✓ | ✓ | ∼ | 98.42 |
| PEPIT-HEBBIAN | OURS | ✓ | ✓ | ✓ | ✓ | ∼ | 98.05 |
| PEPITA-TL | OURS | ✓ | ✓ | ✓ | ✓ | ✓ | 92.78 |

several brain-inspired algorithms have been shown to train ANNs with local information in the frameworks of Predictive Coding (PC) (Rao & Ballard, 1999; van den Oord et al., 2019; Millidge et al., 2022; Salvatori et al., 2021; Ororbia & Mali, 2019; Ororbia et al., 2020) and neural generative coding (Ororbia & Kifer, 2022; Ororbia & Mali, 2023a). Table 1 reports properties of biological plausibility and accuracy metrics for these algorithms. The FA algorithm (Lillicrap et al., 2016a) replaces the transposed forward weights $W$ in the feedback path with random, fixed (non-learning) weight matrices $F$ to deliver error information to earlier layers, thereby solving the weight transport problem (Fig. 1a) (Lillicrap et al., 2016a). While FA achieves a performance close to BP on simple tasks such as MNIST (LeCun & Cortes, 2010) and CIFAR-10 (Krizhevsky et al., a) with relatively shallow networks, it fails to scale to complex tasks and architectures (Bartunov et al., 2018; Xiao et al., 2018; Moskovitz et al., 2018). Moreover, Hinton & McClelland (1987) proposed the Recirculation algorithm in which the output of an autoencoder is fed back into the network as a "negative" example. Later, Baldi & Sadowski (2018) showed that Recirculation is a special case of adaptive FA. To improve FA performances, Akrout et al. (2019) proposed the Weight-Mirror (WM) algorithm, which is an approach to adjust the feedback weights in FA to improve their agreement, allowing to train neural networks with complex architectures (ResNet-18 and ResNet-50) on complex image recognition tasks (ImageNet).

## 2.2 THEORETICAL ANALYSES

The theoretical study of online learning with 1-hidden-layer neural networks has brought considerable insights to statistical learning theory (Saad & Solla, 1995a;b; Riegler & Biehl, 1995; Goldt et al., 2020; Refinetti et al., 2021b). We leverage these works to understand learning with biologically plausible algorithms. Through similar analysis, Refinetti et al. (2021a) analytically confirmed the previous results of Lillicrap et al. (2016b); Nøkland (2016); Frenkel et al. (2021) showing that the key to learning with direct feedback alignment (DFA) is the alignment between the network's weights and the feedback matrices, which allows for the DFA gradient to be aligned to the BP gradient. The authors further show that the fixed nature of the feedback matrices induces a degeneracy-breaking effect where, out of many equally good solutions, a network trained with DFA converges to the one that maximizes the alignment between feedforward and feedback weights. This effect, however, imposes constraints on the structure of the feedback weights for learning, and possibly explains the difficulty of training convolutional neural networks with DFA. Bordelon & Pehlevan (2022) derived self-consistent equations for the learning curves of DFA and FA in infinite-width networks via a path integral formulation of the training dynamics and investigated the impact of biologically plausible rules on feature learning.

## 2.3 THE PEPITA LEARNING RULE

Given a fully connected network with $L$ layers, an input $x$, and one-hot encoded labels $y$ (Fig. 1b), the learning rule proposed by Dellaferrera & Kreiman (2022) relies on a *clean* and a *modulated*

forward pass through the network. During the *clean* pass, the hidden unit and output unit activations are computed as:

$$h_1 = \sigma_1(W_1 x), \quad h_\ell = \sigma_\ell(W_\ell h_{\ell-1}) \text{ for } 2 \leq \ell \leq L, \tag{1}$$

where $\sigma_l$ is the non-linearity at the output of the $\ell^{th}$ layer and $W_\ell$ is the matrix of weights between layers $\ell - 1$ and $\ell$. During the *modulated* pass, the activations are computed as:

$$h_1^{err} = \sigma_1(W_1(x - Fe)), \quad h_\ell^{err} = \sigma_\ell(W_\ell h_{\ell-1}^{err}) \text{ for } 2 \leq \ell \leq L, \tag{2}$$

where $y$ is the target output, $e \equiv h_L - y$ denotes the network error, and $F$ is the fixed random matrix used to project the error on the input. We denote the output of the network either as $h_L$ or $\hat{y}$. After the two forward passes, the weights are updated according to the PEPITA learning rule:

$$\Delta W_1 = (h_1 - h_1^{err})(x - Fe)^\top; \tag{3}$$

$$\Delta W_\ell = (h_\ell - h_\ell^{err})(h_{\ell-1}^{err})^\top \text{ for } 2 \leq \ell < L; \tag{4}$$

$$\Delta W_L = e(h_{L-1}^{err})^\top. \tag{5}$$

Note that, compared to the original paper, we changed the sign convention to read $-Fe$ rather than $+Fe$ in eqn. 2 and eqn. 3. Because the distribution of $F$ entries is symmetric around zero, this has no consequence on the results. This change will appear useful when describing the WM applied to PEPITA in Section 4. Finally, the updates are applied depending on the chosen optimizer. For example, using stochastic gradient descent with learning rate $\eta$: $W(t + 1) = W(t) - \eta \Delta W$. The pseudocode detailing the algorithm is provided in Appendix A.

## 2.4 THE FORWARD-FORWARD ALGORITHM

Analogously to PEPITA, the FF algorithm removes the need for a backward pass and relies on two forward passes (Fig. 1d). The *clean* pass and the *modulated* pass operate on real data and on appropriately distorted data, respectively. The latter is generated to exhibit very different long-range correlations but very similar short-range correlations. In practice, the *modulated* samples are hybrid images obtained by adding together one digit image times a mask with large regions of ones and zeros and a different digit image times the reverse of the mask. In the *clean* pass, the weights are updated to increase a "goodness" in the hidden layers (Hinton (2022) proposes to use the sum of the squared activities, or its negation), and in the *modulated* pass to decrease it. Similarly, the Predictive Forward Forward (PFF) integrates the local synaptic adaptation rule of FF based on a goodness measure and contrastive learning, with lateral competition and aspects of PC, such as its local error Hebbian manner of adjusting generative synaptic weights (Ororbia & Mali, 2023b).

## 3 THEORETICAL ANALYSIS OF THE LEARNING DYNAMICS OF PEPITA

In this section, we present a theory for the "forward-only" learning frameworks, focusing specifically on PEPITA in two-layer networks. We propose a useful approximation of the PEPITA update, that we exploit to derive analytic expressions for the learning curves, and investigate its learning mechanisms in a prototype teacher-student setup.

### 3.1 TAYLOR EXPANSION AND ADAPTIVE FEEDBACK RULE

First, we observe that the perturbation applied in the modulation pass is small compared to the input: $\|Fe\| \ll \|x\|$. Indeed, in the experiments, the entries of the feedback matrix $F$ are drawn with standard deviation $\sigma_F = \kappa/\sqrt{D}$, where $D$ is the input dimension – typically large – and $\kappa$ is a constant set by grid search, while the input entries are of order one. Thus, it is reasonable to Taylor-expand the presynaptic term $h_1 - h_1^{err}$, which results in the approximate update rule:

$$\Delta W_1 \simeq [(W_1 Fe) \odot h_1'] x^\top, \tag{6}$$

where we have used $x$ instead of $(x - Fe)$ since the small perturbation has been found to be negligible for the performance (Dellaferrera & Kreiman, 2022). Eqn. 6 shows that PEPITA is effectively implementing a "DFA-like" update (equivalent to FA in two-layer networks), but using an AF matrix where the random term is modulated by the network weights. In light of this observation, it is natural to expect the alignment between $W_1 F$ and $W_2$. This simple approximation, which we call

Adaptive Feedback Alignment (AFA), is actually very accurate, as we verify numerically in Fig. 2a, displaying experiments on MNIST, and in Fig. S3 of Appendix D for the CIFAR-10 and CIFAR-100 (Krizhevsky et al., b) datasets. AFA approximates PEPITA very accurately also in the teacher-student regression task depicted in Fig. 2b, analyzed in the next section.

### 3.2 ORDINARY DIFFERENTIAL EQUATIONS FOR ADAPTIVE FEEDBACK ALIGNMENT WITH ONLINE LEARNING FOR TEACHER-STUDENT REGRESSION

To proceed in our theoretical analysis, it is useful to assume a generative model for the data. We focus on the classic teacher-student setup (Gardner & Derrida, 1989; Seung et al., 1992; Watkin et al., 1993; Engel & Van den Broeck, 2001; Zdeborová & Krzakala, 2016). We consider $D-$dimensional standard Gaussian input vectors $x \sim \mathcal{N}(0, I_D)$, while the corresponding label $y = \tilde{W}_2 \, \tilde{\sigma}(\tilde{W}_1 x)$ is generated by a two-layer *teacher* network with fixed random weights $\tilde{W}_1$, $\tilde{W}_2$ and activation function $\tilde{\sigma}(\cdot)$. The two-layer *student* network outputs a prediction $\hat{y} = W_2 \, \sigma(W_1 x)$ and is trained with the AFA rule and an *online* (or *one-pass*) protocol, i.e., employing a previously unseen example $x_\mu, \mu = 1, \ldots, N$, at each training step, where $N$ is the number of samples. We characterize the dynamics of the mean-squared generalization error

$$\epsilon_g \equiv \frac{1}{2}\mathbb{E}_x\big[(\hat{y} - y)^2\big] \equiv \frac{1}{2}\mathbb{E}_x\big[e^2\big], \quad e \equiv \hat{y} - y, \tag{7}$$

in the infinite-dimensional limit of both input dimension $D \to \infty$ and number of samples $N \to \infty$, at a finite rate (or *sample complexity*) $N/D \sim \mathcal{O}_D(1)$ where the training time is $t = \mu/D$. The hidden-layer size is of order $\mathcal{O}(1)$ in both teacher and student. We follow the derivation in the seminal works of Biehl & Schwarze (1995); Saad & Solla (1995d;c), which has been put on rigorous ground by Goldt et al. (2019), and extend it to include the time-evolution of the AF. As discussed in Appendix E, the dynamics of the error $\epsilon_g$ as a function of training time $t$ is fully captured by the evolution of the AF matrix and a set of low-dimensional matrices encoding the teacher-student alignment. We derive a closed set of ordinary differential equations (ODEs) tracking these matrices, and we integrate them to obtain our theoretical predictions. Furthermore, in Appendix E.1, we perform an expansion at early training times $t \ll 1$ to elucidate the alignment mechanism and the importance of the "teacher-feedback alignment", i.e. $\tilde{W}_2\tilde{W}_1 F$, as well as the norm of $\|\tilde{W}_1 F\|$. For the sake of the discussion, we consider sigmoidal activations $\tilde{\sigma}(\cdot) = \sigma(\cdot) = \mathrm{erf}(\cdot)$, keeping in mind that the symmetry $\mathrm{erf}(-x) = -\mathrm{erf}(x)$ induces a degeneracy of solutions.

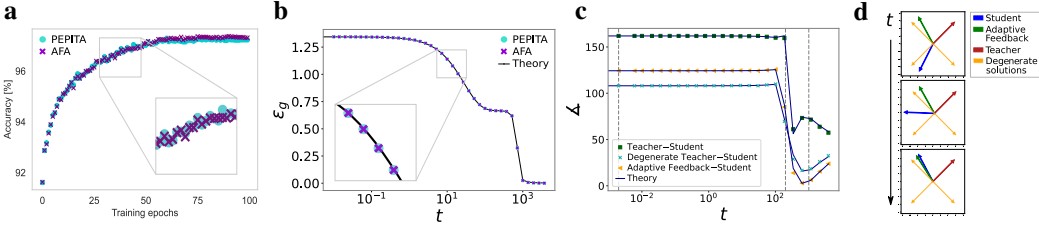

Figure 2: **(a)** Test accuracy as a function of epochs for experiments with the MNIST and a 1-hidden-layer network with (1024 hidden units, ReLU). Blue dots mark the "vanilla" PEPITA algorithm (without momentum) while purple crosses mark the AFA approximation (6). **(b)** Generalization error as a function of time for the experiments with PEPITA (blue dots) and AFA (purple crosses), and the theoretical curves (App. E) marked by full lines. Parameters: $D = 500$, $lr = .05$, erf activation, 2 hidden units in both teacher and student. **(c)** Alignment angle between the teacher and student second-layer weights (dark green), the student and a degenerate solution (light green), and the AF matrix and the student (orange) as a function of time. **(d)** Direction of the student, the AF and the teacher (and degenerate solutions). Different time shots are marked by vertical dashed lines in panel (c).

Fig. 2b shows our theoretical prediction for the generalization error as a function of time, compared to numerical simulations of the PEPITA and AFA algorithms. We find excellent agreement between our infinite-dimensional theory and experiments, even at moderate system size ($D = 500$).

Fig. 2c shows the dynamic changes in the alignment of the second-layer student weights $W_2$ with different matrices: the second-layer teacher weights $\tilde{W}_2$, the second-layer teacher weights of the

closest degenerate solution, and the AF weights $W_1 F$. We clearly observe that the error is stuck at a plateau until "adaptive-feedback alignment" happens ($t \sim 10^2$). Fig. 2d depicts the directions of the two-dimensional vectors $W_2$, $W_1 F$, and $\tilde{W}_2$ at three different time shots, marked by vertical dashed lines in Fig. 2c. It is crucial to notice that, in this case, the feedback also evolves in time, giving rise to a richer picture, that we further discuss in Appendix D. In the special case of Fig. 2c, while the direction of $W_1 F$ is almost constant, its norm increases, speeding up the learning process, as shown in Fig. S4. Similar behavior was observed in Dellaferrera & Kreiman (2022). Notice that, in the case of one hidden layer, FA directly implies gradient alignment given that the weight updates become identical if the feedback matches the feed-forward weights.

# 4 TESTING PEPITA ON DEEPER NETWORKS, WITH WEIGHT MIRRORING, WEIGHT DECAY AND ACTIVATION NORMALIZATION

The theoretical analysis above focuses on a two-layer network, as used in the proof-of-principle demonstration of PEPITA's function in the original article. To the best of our knowledge, PEPITA has been tested so far only on two-layer networks. Here, we empirically show that the algorithm can be extended to train deeper networks. Furthermore, we propose three techniques that improve the performance of PEPITA and narrow the gap with BP. First, we enhance PEPITA with standard techniques used in ML, namely weight decay (WD) (Krogh & Hertz, 1991) and activation normalization (as in Hinton, 2022). In addition, we combine PEPITA with the WM algorithm (Akrout et al., 2019) to train the projection matrix $F$. Indeed, WM has been shown to greatly improve alignment between the forward and backward connections and the consequent accuracy of FA and, in Section 3.1, we have shown that PEPITA is closely related to FA. In order to apply WM to PEPITA, we propose a generalization of WM for learning rules where the dimensionalities of the feedback and feedforward weights do not match, such as PEPITA, but also including DFA. In our work we focus on fully connected models, and leave the enhancement of convolutional networks for further exploration. Experiments were run on Nvidia V100 GPUs, using custom Python code available here.

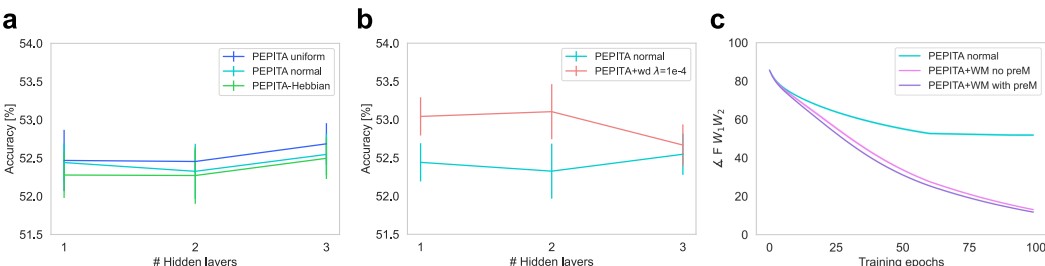

Figure 3: **(a)**, **(b)** Test accuracy of fully connected networks with increasing depth trained with PEPITA on the CIFAR-10 dataset. **(a)** PEPITA uniform and PEPITA normal refer to the initialization of the weights and $F$ (Sec. 4). "PEPITA-Hebbian" refers to the learning rule explained in Section 5.1. **(b)** Effect of weight decay. **(c)** Alignment angle between $F$ (Eq. 2) and $W_1 \cdot W_2$ during training with or without WM. PreM refers to pre-mirroring (Sec. 4). Hyperparameters are reported in Table S3. The plots indicate mean and standard deviation over 10 independent runs.

## 4.1 METHODS

We first test two initialization strategies, He uniform (as in Dellaferrera & Kreiman, 2022) and He normal. The normal initialization is necessary to factorize the feedback matrix $F$ as explained below. The distribution of the entries of F has a mean of zero and standard deviation reported in Tab. S3. When not specified, we use the He normal initialization. The hyperparameters are found through grid search and are reported in Table S3.

In order to apply WM to PEPITA, we need to modify the original algorithm (Akrout et al., 2019). Indeed, in a network trained with FA, the WM algorithm aligns, for each layer $\ell$, a backward matrix $F_\ell$ with the transpose of the corresponding forward matrix $W_\ell$ (Fig. 1a). In order to recover the one-to-one correspondence of WM, we factorize the $F$ matrix into the product of as many matrices

as the number of layers: $F = \prod_{\ell=1}^{L} F_\ell$ (Fig. 1c). We initialize each of these matrices by sampling the entries from a normal distribution and apply WM layer-wise (see pseudocode in Appendix J). After each standard WM update, we also normalize each $F_\ell$ to keep the standard deviation $\sigma_F^{(t)}$ of the updated $F$ at step $t$ constant at its initialization value $\sigma_F^{(0)}$ (see Table S3). Running WM before learning starts (*pre-mirroring*) brings the random feedback weights into alignment without the use of any training data. We run benchmarks both with five epochs of pre-mirroring, and without. We compared PEPITA's performance to three baselines, BP, FA, and Direct Random Target Projection (DRTP, Frenkel et al., 2021) with and without weight decay. For the baselines, we use the code by Frenkel et al. (2021). Our use of learning rate decay means the grid search returned different hyperparameters, explaining some discrepancies in the accuracies compared to Frenkel et al. (2021).

Table 2: Test accuracy [%] achieved by BP, FA, DRTP, and PEPITA in the experiments, with and without weight decay, normalization of the activations and WM . Mean and standard deviation are computed over 10 independent runs. All training schemes are tested with ReLU nonlinearity except DRTP which uses tanh nonlinearity. The hyperparameters obtained through grid search are reported in Table S3. Bold fonts refer to the best results exclusively among PEPITA and its improvements.

| | W. Decay | Norm. | Mirror | MNIST | CIFAR-10 | CIFAR-100 |
|---|---|---|---|---|---|---|
| BP | ✗ | ✗ | ✗ | 98.72±0.06 | 57.60±0.20 | 29.36±0.19 |
| | ✓ | ✗ | ✗ | 98.66±0.04 | 57.92±0.19 | 29.93±0.27 |
| FA | ✗ | ✗ | ✗ | 98.48±0.05 | 56.76±0.23 | 22.75± 0.28 |
| | ✓ | ✗ | ✗ | 98.35±0.04 | 57.19±0.24 | 22.62±0.25 |
| DRTP | ✗ | ✗ | ✗ | 95.45±0.09 | 46.92±0.21 | 16.99±0.16 |
| | ✓ | ✗ | ✗ | 95.44±0.09 | 46.92±0.27 | 17.59±0.18 |
| PEPITA | ✗ | ✗ | ✗ | 98.02±0.08 | 52.45±0.25 | 24.69±0.17 |
| | ✓ | ✗ | ✗ | 98.12±0.08 | 53.05±0.23 | 24.86±0.18 |
| | ✗ | ✓ | ✗ | **98.41**±0.08 | **53.51**±0.23 | 22.87±0.25 |
| | ✗ | ✗ | ✓ | 98.05±0.08 | 52.63±0.30 | **27.07**±0.11 |
| | ✓ | ✗ | ✓ | 98.10±0.12 | 53.46±0.26 | **27.04**±0.19 |
| | ✗ | ✓ | ✓ | **98.42**±0.05 | **53.80**±0.25 | 24.20±0.36 |

## 4.2 Results

**Initialization** The He uniform and normal initialization performed equally (Fig. 3a).

**Depth** Without activation normalization, PEPITA is successful in training networks with up to three hidden layers without drop in accuracy for increasing depth (Fig. 3a). Furthermore, with activation normalization, PEPITA can train networks with up to five hidden layers. However, this comes at the cost of a decrease in performance with increasing depth (Fig. S5).

**Weight decay and normalization** On a 1-hidden-layer network trained on the CIFAR-10 dataset, weight decay and normalization improved PEPITA's performance by 0.6% and 1.1%, respectively (Fig. 3b and Table 2).

**Weight mirroring** Fig. 3c shows that, when WM is applied, the $W$ and $F$ matrices achieve a significantly better alignment than with the standard PEPITA. The improvement of the alignment is more significant for 1- and 2-hidden-layer than for 3-hidden-layer networks (Sup. Fig. S8). Furthermore, the improved alignment obtained with WM is reflected in improved accuracy, especially for 1-hidden-layer networks (Table 2), but also for 2- and 3-hidden-layer networks (Sup. Table S1). Remarkably, on the CIFAR-10 dataset, WM combined with activation normalization leads to 53.8%, an improvement of 1.3% compared to the standard PEPITA. On CIFAR-100 WM improves the accuracy by over 2%, reducing significantly the gap with BP.

**Convergence rate** Dellaferrera & Kreiman (2022) shows that PEPITA's convergence rate is in between BP (the fastest) and FA (the slowest). By extracting the *slowness* from the *plateau equation for learning curves* (Dellaferrera et al., 2022), we find that pre-mirroring improves the convergence rate for 1-hidden-layer models (Table S2), compatibly with the "align, then memorize" paradigm proposed for FA (Refinetti et al., 2021a).

## 5    ON THE RELATIONSHIP BETWEEN FORWARD-FORWARD AND PEPITA

The goal of this section is to compare PEPITA and a FF framework with top-down connections. To this end, we first propose a minor change to the PEPITA update rule that exposes its Hebbian foundation, and which can be applied to train ANNs in a time-local fashion. Finally we analytically compare FF and PEPITA in the case where negative samples are based on high-level feedback.

### 5.1    PEPITA'S FORMULATION AS HEBBIAN AND ANTI-HEBBIAN PHASES

In the PEPITA framework, the presynaptic term ($h_{\ell-1}^{err}$) in the learning rule of eqn. 4 can be replaced interchangeably with the same term, computed during the first forward pass ($h_{\ell-1}$) (Dellaferrera & Kreiman, 2022). Taking this a step further, we separate the brackets in eqn. 4 and mix the two choices for the presynaptic term obtaining an approximately equivalent rule for hidden layer weights:

$$\Delta W_\ell = h_\ell h_{\ell-1}^{err\top} - h_\ell^{err} h_{\ell-1}^{err\top} \simeq \quad h_\ell h_{\ell-1}^\top - h_\ell^{err} h_{\ell-1}^{err\top}. \tag{8}$$

The learning rule eqn. 8 now contains two Hebbian terms, each a product of the activity of the presynaptic and the postsynaptic node, similarly to (Xie & Seung, 2003; Scellier & Bengio, 2017; Detorakis et al., 2019). We dub this approximation *PEPITA-Hebbian*. Fig. 3a shows that the approximation in eqn. 8 has negligible impact on the accuracy. We note that these two terms can be used to update the weights *online*, separating the updates between the first and second pass (Algorithm S2), and allowing to alleviate the constraint of storing in memory information of the first pass until the second pass. We name this version *PEPITA-time-local* (PEPITA-TL). PEPITA-TL exhibits a decrease in accuracy (Sup. Fig. S1), but it is still uses the feedback in a useful manner (Appendix B). We leave the exploration to improve the accuracy of PEPITA-TL for further work.

### 5.2    COMPARISON BETWEEN THE WEIGHT UPDATES OF PEPITA AND FORWARD-FORWARD

In FF, the weights are updated to increase and decrease the "goodness" in every hidden layer in the *clean* pass and *modulated* pass respectively, where the goodness can be the negative sum of the squared neural activities (in our notation, $-\|h_\ell\|^2$, for the *clean* and $-\|h_\ell^{err}\|^2$ for the *modulated* pass). Hinton (2022) chooses a loss based on the logistic function $\sigma$ applied to the goodness, minus a threshold, $\theta$: $p = \sigma\left(\|h_l\|^2 - \theta\right)$. For simplicity, we work directly on the goodness. This is equivalent to minimizing, at each layer $\ell$, a loss function defined as:

$$J_\ell = \|h_\ell\|^2 - \|h_\ell^{err}\|^2 \tag{9}$$

We then compute an update for FF as the derivative of the loss eqn. 9 with respect to the weights:

$$\frac{1}{2}\frac{\partial J_\ell}{\partial W_\ell} = \frac{1}{2}\left(\frac{\partial\|h_\ell\|^2}{\partial W_\ell} - \frac{\partial\|h_\ell^{err}\|^2}{\partial W_\ell}\right)$$
$$= \frac{1}{2}\left(\frac{\partial\|\sigma(W_\ell h_{\ell-1})\|^2}{\partial W_\ell} - \frac{\partial\|\sigma(W_\ell h_{\ell-1}^{err})\|^2}{\partial W_\ell}\right)$$
$$= \sigma(W_\ell h_{\ell-1}) \odot \sigma'(W_\ell h_{\ell-1})h_{\ell-1}^\top - \sigma(W_\ell h_{\ell-1}^{err}) \odot \sigma'(W_\ell h_{\ell-1}^{err})h_{\ell-1}^{err\top}$$
$$= (\sigma'(W_\ell h_{\ell-1}) \odot h_\ell h_{\ell-1}^\top - \sigma'(W_\ell h_{\ell-1}^{err}) \odot h_\ell^{err} h_{\ell-1}^{err})$$
$$= (h_\ell' \odot h_\ell)h_{\ell-1}^\top - (h_\ell^{err'} \odot h_\ell^{err})h_{\ell-1}^{err\top}.$$

We note that the terms $h_\ell' \equiv \sigma'(W_\ell h_{\ell-1})$ and $h_\ell^{err'} \equiv \sigma'(W_\ell h_{\ell-1}^{err})$ can be omitted in the common case where $\sigma$ is a ReLU function because they are 0 if and only if $h_\ell$ is already 0, and equal to 1 otherwise. Likewise, they are always equal to 1 if $\sigma$ is linear. In these cases, therefore

$$\frac{1}{2}\frac{\partial J_\ell}{\partial W_\ell} = h_\ell h_{\ell-1}^\top - h_\ell^{err} h_{\ell-1}^{err\top}, \tag{10}$$

which is equivalent to eqn. 8, apart from a factor $1/2$ that can be incorporated in the learning rate. For other choices of $\sigma$, since $Fe \ll x$, the derivative of the activations in the two forward passes are close enough to approximate $\sigma'(W_\ell h_{\ell-1}) \simeq \sigma'(W_\ell h_{\ell-1}^{err})$.

By comparing eqn. 8 and eqn. 10, we observe that the FF update rule with loss eqn. 9 approximates the weight update rule of the previously proposed PEPITA. We remark that the comparison between

PEPITA and FF holds for the general FF framework minimizing and maximizing the square of the activities for the clean and modulated passes respectively, which in the experiments by Hinton (2022) provides the best performances. However, the weight update reported by Hinton (2022) presents analytical differences from eqn. 10 due to their use of thresholding and non-linearity applied to the goodness. Another notable difference lies in the way the "modulated" samples are generated in the two algorithms: as described, in PEPITA's *modulated* pass, the input is modulated by the error. In the FF algorithm, in contrast, the negative samples are data vectors corrupted by external means, although Hinton (2022) mentions that the negative data may be predicted by the neural net using top-down connections. In contrast, the activations of the two passes are not statistically different in PEPITA, presumably due to the small perturbation of the input in the second pass (Fig. S2).

## 6 DISCUSSION

In the quest for biologically inspired learning mechanisms, two of the long-standing challenges include providing a theoretical ground to understand the dynamics of the training algorithms and scaling their application to complex networks and datasets. The FF algorithm (Hinton, 2022) and the PEPITA algorithm (Dellaferrera & Kreiman, 2022) are "forward-only" algorithms that train neural networks with local information, without weight transport, without freezing the network's activity, and without backward locking.

First, we show that PEPITA can be approximated by an FA-like algorithm, where the error is propagated to the input layer via the project matrix $F$ "adapted" through the forward synaptic matrix. By performing a theoretical characterization of the generalization dynamics, we provide intuition on the alignment mechanisms, offering novel theoretical insights into the family of forward-only algorithms. We then test two standard techniques to improve the performance of PEPITA, i.e. weight decay, and activation normalization. Moreover, in line with our theoretical findings that highlight the importance of the alignment of the feedback and feedforward weights, we test PEPITA in combination with a version of WM tailored to PEPITA. By aligning the feedback weights to the feedforward weights, WM significantly improves the classification accuracy on both the CIFAR-10 and CIFAR-100 datasets. Finally, we propose *PEPITA-hebbian*, a version of PEPITA in which the update contains a sum of Hebbian and anti-Hebbian terms. We compare *PEPITA-hebbian* to FF, highlighting the similarities between the two algorithms. While FF uses externally generated negative samples for the second forward pass, Hinton (2022) himself suggests the possibility that the negative data may be predicted by the neural net using top-down connections. We show that, when choosing the goodness as the negative sum of the squared activities, *PEPITA-hebbian* is precisely an example of this possibility. This avoids the biologically unrealistic requirement of corrupting the images entailed by FF and is compatible with biological mechanisms of neuromodulation and thalamocortical projections (Mazzoni et al., 1991; Williams, 1992; Clark et al., 2021; Kreiman, 2021). Moroever, the separate feedback pathway can be linked to neuromodulatory signals that acts as *plasticity-steering* effects (Roelfsema & Holtmaat, 2018), necessary for PEPITA-TL to distinguish between the two forward passes. We envision that further research could build a unified theoretical framework for these and other forward-only algorithms, leading to higher accuracy, network depth, and biological plausibility.

## 7 LIMITATIONS AND FUTURE WORK

While this work provides valuable insights into forward-learning algorithms from empirical and analytical perspectives, it is important to acknowledge several limitations. First, the algorithms we analyze still lag behind BP in terms of performance and scalability. Further work is needed to understand the causes of the difference in accuracy between forward-learning algorithms and BP, which may include poor collaboration between layers during training (Lorberbom et al., 2023). Moreover, the factorization of the feedback matrix in the modified version of WM may not correspond to any known biological mechanism. Additional, more bio-plausible alternatives need to be explored. Another open direction of investigation is the theoretical analysis of the learning dynamics in deep networks. A starting point in this direction could be to consider deeper networks. In conclusion, our work provides a starting point for addressing those limitations through further research.

## ACKNOWLEDGEMENTS

This work was supported by NIH grant R01EY026025 and NSF grant CCF-1231216. M.S. was supported by an ETH AI Center postdoctoral fellowship.

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

## A   PSEUDOCODE FOR PEPITA

Algorithm S1 describes the original PEPITA as presented in Dellaferrera & Kreiman (2022). Algorithm S2 describes our modification of PEPITA, which is Hebbian and local both in space and in time (Pepita-time-local).

---

**Algorithm S1** Implementation of PEPITA

**Require:** Input $x$ and one-hot encoded label $y$
  {standard forward pass}
  $h_0 = x$
  **for** $\ell = 1, ..., L$ **do**
    $h_\ell = \sigma_\ell(W_\ell h_{\ell-1})$
  **end for**
  $e = h_L - y$
  {modulated forward pass}
  $h_0^{err} = x + Fe$
  **for** $\ell = 1, ..., L$ **do**
    $h_\ell^{err} = \sigma_\ell(W_\ell h_{\ell-1}^{err})$
    **if** $\ell < L$ **then**
      $\Delta W_\ell = (h_\ell - h_\ell^{err})(h_{\ell-1}^{err})^\top$
    **else**
      $\Delta W_\ell = e(h_{\ell-1}^{err})^\top$
    **end if**
    $W_\ell(t+1) = W_\ell(t) - \eta\Delta W_\ell$ {apply update}
  **end for**

---

**Algorithm S2** Impl. of PEPITA-time-local

**Require:** Input $x$ and one-hot encoded label $y$
  {standard forward pass}
  $h_0 = x$
  **for** $\ell = 1, ..., L$ **do**
    $h_\ell = \sigma_\ell(W_\ell h_{\ell-1})$
    $\Delta W_\ell^+ = h_\ell h_{\ell-1}^\top$
    $W_\ell^+(t+1) = W_\ell(t) - \eta\Delta W_\ell^+$ {apply 1st update}
  **end for**
  $e = h_L - y$
  {modulated forward pass}
  $h_0^{err} = x - Fe$
  **for** $\ell = 1, ..., L$ **do**
    $h_\ell^{err} = \sigma_\ell(W_\ell^+ h_{\ell-1}^{err})$
    **if** $\ell < L$ **then**
      $\Delta W_\ell^- = -h_\ell^{err} h_{\ell-1}^{err\top}$
    **else**
      $\Delta W_\ell^- = -y h_{\ell-1}^{err\top}$
    **end if**
    $W_\ell(t+1) = W_\ell^+(t+1) - \eta\Delta W_\ell^-$ {apply 2nd update}
  **end for**

---

The updates for PEPITA-Hebbian are:

- for the hidden layers:

$$\begin{aligned}
\Delta W_\ell &= h_\ell h_{\ell-1}^{err\top} - h_\ell^{err} h_{\ell-1}^{err\top} \\
&\simeq h_\ell h_{\ell-1}^\top - h_\ell^{err} h_{\ell-1}^{err\top},
\end{aligned} \tag{11}$$

- for the first and last layers

$$\begin{aligned}
\Delta W_1 &\simeq h_1 x^\top - h_1^{err}(x - Fe)^\top; \\
\Delta W_L &\simeq h_L h_{L-1}^\top - y h_{L-1}^{err\top}.
\end{aligned} \tag{12}$$

These updates are applied at the end of both forward passes for PEPITA-Hebbian, similarly as in pseudocode S1.

# B   TRAINING WITH THE TIME-LOCAL RULE

The weight update of the *PEPITA-Hebbian* rule in eqn. 8 consists of two separate phases that can contribute to learning without knowledge of the activity of the other, i.e. learning is time-local. Specifically, the term $h_\ell h_{\ell-1}^\top$ can be applied *online*, immediately as the activations of the first pass are computed. Analogously, the second term $-h_\ell^{err} h_{\ell-1}^{err\top}$ can be applied immediately during the second forward pass. To ensure that the hidden-layer updates prescribed by PEPITA are useful, we compared the test curve of PEPITA-TL against a control with $F = 0$ (Fig. S1), and found that removing the feedback decreases the accuracy from approx. 40% to approx. 18%.

Regarding the time-locality of FF, the two forward passes can be computed in parallel, as the *modulated* pass does not need to wait for the computation of the error of the first pass. However, according to the available implementations (Mukherjee, 2023) the updates related to both passes are applied together at the end of the second forward pass.

The time-local PEPITA is trained on the CIFAR-10 dataset with learning rate 0.01. All the other hyperparameters are the same as the ones reported in Table S3.

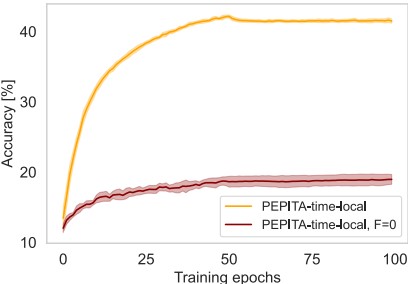

Figure S1: Test curve for PEPITA in its time-local formulation and time-local PEPITA with F=0 (i.e., only the last layer is trained) on the CIFAR-10 dataset. The network has 1 hidden layer with 1024 units. The forward matrices are initialized using the He normal initialization. $F$ entries are sampled from a normal distribution with standard deviation $0.5 \cdot 2\sqrt{6/(32 \cdot 32 \cdot 3)}$. We use learning rate 0.0001 and weight decay with $\lambda = 10^{-4}$. The learning is reduced by a factor of $\times 0.1$ at epoch 50. The plot indicates mean and standard deviation over 10 independent runs. Time-local PEPITA achieves a significantly higher accuracy than the time-local, F=0 scheme.

## C   DISTRIBUTION OF THE GOODNESS IN PEPITA

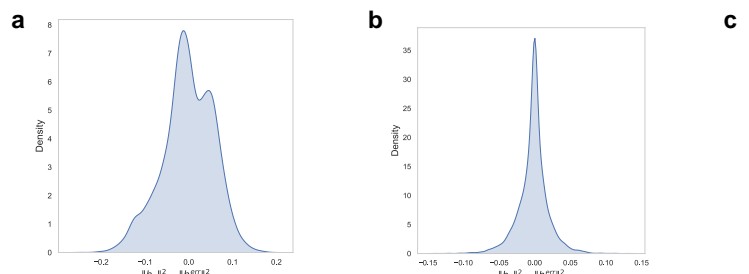

Figure S2: Difference of the norm of the squared activities of the first hidden layer between the *clean* and *modulated* pass in PEPITA (a) before training, (b) after 50 epochs, and (c) at the end of training. The network is a 2-hidden-layer network trained with WD with $\lambda = 10^{-4}$ on the CIFAR-10 dataset. The activites are recorded on the test set. We remark that in PEPITA the input of the second forward pass is modulated by the error. Since the error decreases during training, also the difference of the activations in the two passes decreases with training. This explains why the distribution of the difference of the norm of the squared activities has a lower standard deviation in the middle of training (b) and at the end of training (c), than before training (a). In contrast, the modulation of the input in FF is constant during training, and the scope of training is maximising the difference of the goodness in the two passes.

## D   Additional figures on the Adaptive Feedback Alignment approximation

Fig. S3 displays the comparison between the "vanilla" PEPITA algorithm and the AFA approximation introduced in eqn. 6 of the main text. The test accuracy as a function of training epochs is depicted for the CIFAR-10 dataset in the left panel and for the CIFAR-100 dataset in the right panel.

Fig. S4 depicts the norm of the AF matrix $f = W_1 F / D$ as a function of training time. The symbols mark the numerical simulations at dimension $D = 500$, while the full line represents our theoretical prediction. We observe that, for this run, the norm of the AF increases over time. We have observed by numerical inspection that this behavior is crucial to speed up the dynamics, as also observed in Dellaferrera & Kreiman (2022).

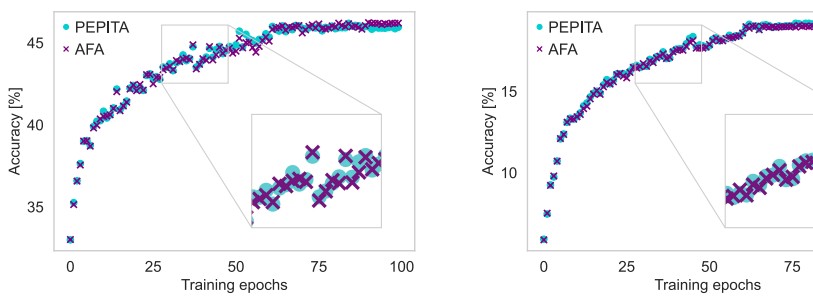

Figure S3: Comparison between the test accuracy as a function of training epochs between the "vanilla" PEPITA algorithm and its AFA approximation (eqn. 6 of the main text) for the CIFAR-10 (*left panel*) and CIFAR-100 (*right panel*) datasets.

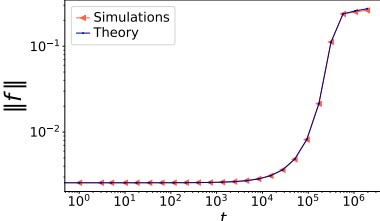

Figure S4: Norm of the alignment matrix $f = W_1 F / D$ as a function of training time, for the same parameters as in Fig. 2 of the main text: $D = 500$, $lr = .05$, erf activation, two hidden units in both teacher and student ($K = M = 2$).

## E   Ordinary differential equations for online learning in the teacher-student regression task

In this section, we present the details of the teacher-student model under consideration and we sketch the derivation of the ordinary differential equations tracking the online learning dynamics of the AFA rule. We consider a shallow *student* network trained with AFA to solve a supervised learning task. The input data are random $D$−dimensional vectors $x \in \mathbb{R}^D$ with independent identically distributed (i.i.d.) standard Gaussian entries $x_j \sim \mathcal{N}(0, 1)$, $j = 1, \ldots, D$, and the (scalar) labels are generated as the output of a 1-hidden-layer *teacher* network with parameters $\tilde{\theta} = (\tilde{W}_1, \tilde{W}_2, M, \tilde{\sigma})$:

$$y = \sum_{m=1}^{M} \tilde{W}_2^m \tilde{\sigma}(\nu^m), \qquad \nu^m = \frac{\tilde{W}_1^m x}{\sqrt{D}}, \tag{13}$$

where $M$ is the size of the teacher hidden layer, $\nu^m$ denotes the teacher preactivation at unit $m \in \{1, \ldots, M\}$, and $\tilde{\sigma}(\cdot)$ is the activation function. The student is a 1-hidden-layer neural network parametrized by $\theta = (W_1, W_2, K, \sigma)$ that outputs the prediction

$$\hat{y} = \sum_{k=1}^{K} W_2^k \sigma(\lambda^k), \qquad \lambda^k = \frac{W_1^k x}{\sqrt{D}}, \tag{14}$$

where $K$ is the size of the student hidden layer, $\sigma(\cdot)$ the student activation function, $\lambda^k$ the student preactivation at unit $k \in \{1, \ldots, K\}$. For future convenience, we write explicitly the scaling with respect to the input dimension. Therefore, at variance with the main text, in this supplementary section we always rescale the first layer weights as well as the feedback by $1/\sqrt{D}$.

We focus on the *online* (or *one-pass*) learning protocol, so that at each training time the student network is presented with a fresh example $x_\mu$, $\mu = 1, \ldots N$, and $N/D \sim \mathcal{O}_D(1)$. The weights are updated according to the AFA rule defined in eqn. 6:

$$W_1(\mu + 1) = W_1(\mu) - \eta_1 \Delta W_1(\mu), \qquad \Delta W_1 = \frac{W_1 F}{D} e \, h_1' \frac{x^\top}{\sqrt{D}}, \tag{15}$$

$$W_2(\mu + 1) = W_2(\mu) - \eta_2 \Delta W_2(\mu), \qquad \Delta W_2 = e \, h_1^\top. \tag{16}$$

We consider fixed learning rates $\eta_1 = \eta$, $\eta_2 = \eta/D$. Different learning rate regimes have been explored in Veiga et al. (2022). It is crucial to notice that the mean squared generalization error

$$\epsilon_g(\theta, \tilde{\theta}) = \frac{1}{2} \mathbb{E}_x \left[ \left( \sum_{k=1}^{K} W_2^k \sigma(\lambda^k) - \sum_{m=1}^{M} \tilde{W}_2^m \tilde{\sigma}(\nu^m) \right)^2 \right] \tag{17}$$

depends on the high-dimensional input expectation only through the low-dimensional expectation over the preactivations $\{\lambda_k\}_{k=1}^{K}, \{\nu_m\}_{m=1}^{M}$. Notice that, in this online-learning setup, the input $x$ is independent of the weights, which are held fixed when taking the expectation. Furthermore, due to the Gaussianity of the inputs, the preactivations are also jointly Gaussian with zero mean and second moments:

$$Q^{kl} = \mathbb{E}_x \left[ \lambda^k \lambda^l \right] = \frac{W_1^k \cdot W_1^l}{D}, \tag{18}$$

$$R^{km} = \mathbb{E}_x \left[ \lambda^k \nu^m \right] = \frac{W_1^k \cdot \tilde{W}_1^m}{D}, \tag{19}$$

$$T^{mn} = \mathbb{E}_x \left[ \nu^m \nu^n \right] = \frac{\tilde{W}_1^m \cdot \tilde{W}_1^n}{D}. \tag{20}$$

The above matrices are named *order parameters* in the statistical physics literature and play an important role in the interpretation. The matrices $Q$ and $T$ capture the self-overlap of the student and teacher networks respectively, while the matrix $R$ encodes the teacher-student overlap. In the infinite-dimensional limit discussed above, the generalization error is only a function of the order parameters $Q, T, R$ and of the second layer weights $\tilde{W}_2, W_2$ of teacher and student respectively. Therefore, by tracking the evolution of these matrices via a set of ODEs – their "equations of motion" – we obtain theoretical predictions for the learning curves. The update equations for $Q, R, W_2$ can be obtained from eqn. 15 according to the following rationale. As an example, we consider the update equation for the matrix $Q$:

$$Q^{kl}(\mu + 1) - Q^{kl}(\mu)$$
$$= \frac{1}{D} \left[ W_1^k(\mu) - \eta \Delta W_1^k(\mu) \right] \cdot \left[ W_1^l(\mu) - \eta \Delta W_1^l(\mu) \right] - \frac{1}{D} W_1^k(\mu) \cdot W_1^l(\mu) \tag{21}$$
$$= -\frac{1}{D} \eta \, f^k \, e \, \sigma'(\lambda^k) \lambda^l - \frac{1}{D} \eta \, f^l \, e \, \sigma'(\lambda^l) \lambda^k + \frac{1}{D} \eta^2 \, f^k \, f^l \, \sigma'(\lambda^k) \sigma'(\lambda^l) e^2,$$

where we have defined the AF $f := W_1 F / D$, we have used that $\|x_\mu\|^2 = D$ as $D \to \infty$ and omitted the $\mu$−dependence on the right hand side for simplicity. By taking $t = \mu/D$, as shown in Goldt et al. (2019), in the infinite-dimensional limit $Q^{kl}(\mu)$ concentrates to the solution of the following ODE:

$$\frac{\mathrm{d}Q^{kl}}{\mathrm{d}t} = -\eta f^k \mathbb{E} \left[ \sigma'(\lambda^k) \lambda^l e \right] - \eta f^l \mathbb{E} \left[ \sigma'(\lambda^l) \lambda^k e \right] + \eta^2 f^k f^l \mathbb{E} \left[ \sigma'(\lambda^k) \sigma'(\lambda^l) e^2 \right], \tag{22}$$

Similarly, we can derive ODEs for the evolution of $R, W_2$ and the AF $f$:

$$\frac{\mathrm{d}R^{km}}{\mathrm{d}t} = -\eta\, f^k \mathbb{E}\left[\sigma'(\lambda^k)\nu^m e\right], \quad \frac{\mathrm{d}W_2^k}{\mathrm{d}t} = -\eta \mathbb{E}\left[\sigma(\lambda^k)e\right], \quad \frac{\mathrm{d}f^k}{\mathrm{d}t} = -\eta f^k \mathbb{E}\left[\rho\sigma'(\lambda^k)e\right], \quad (23)$$

where the expectations are taken over the preactivatons and $\rho = F\, x/\sqrt{D}$, and we have $\mathbb{E}_x[\lambda^k \rho] = f^k$, $\mathbb{E}_x[\nu^m \rho] = \tilde{f}^m := \tilde{W}_1 F/D$, $q_f := F \cdot F/D$. The generalization error can be rewritten as

$$\lim_{D\to\infty} \epsilon_g(\theta, \tilde{\theta}) = \frac{1}{2} \sum_{k,l=1}^{K} W_2^k W_2^l\, I_2(k,l) + \frac{1}{2} \sum_{m,n=1}^{M} \tilde{W}_2^m \tilde{W}_2^n\, I_2(m,n) - \sum_{k=1}^{K} \sum_{m=1}^{M} W_2^k \tilde{W}_2^m\, I_2(k,m),$$
$$(24)$$

where $I_2$ generically encodes the averages over the activations

$$I_2(\alpha, \beta) = \mathbb{E}\left[\sigma_\alpha(\gamma^\alpha)\sigma_\beta(\gamma^\beta)\right], \quad \gamma^\alpha = \begin{cases} \lambda^k & \text{if } \alpha = k,l \\ \nu^m & \text{if } \alpha = m,n \end{cases}, \quad \sigma_\alpha = \begin{cases} \sigma & \text{if } \alpha = k,l \\ \tilde{\sigma} & \text{if } \alpha = m,n \end{cases}. \quad (25)$$

The other averages in eqns. can be expressed in a similar way and estimated by Monte Carlo methods. In the case of sigmoidal activation $\sigma(x) = \mathrm{erf}(x/\sqrt{2})$, the function $I_2$ has an analytic expression.

$$I_2(\alpha, \beta) = \frac{2}{\pi}\arcsin\frac{C^{\alpha\beta}}{\sqrt{1 + C^{\alpha\alpha}}\sqrt{1 + C^{\beta\beta}}}, \quad C^{kl} = Q^{kl},\ C^{km} = R^{km},\ C^{mn} = T^{mn}. \quad (26)$$

### E.1 EARLY-TRAINING EXPANSION

As done by Refinetti et al. (2021a) for DFA, it is instructive to consider an expansion of the ODEs at early training times. We assume the following initialization: $W_2^k(0) = 0$, $\forall k \in \{1, \ldots, K\}$, while the first layer is assumed to be orthogonal to the teacher $W_1^k(0) \cdot \tilde{W}_1^m = 0$ and of fixed norm $\|W_1^k(0)\|^2/D = q_0$, $\forall k \in \{1, \ldots, K\}, \forall m \in \{1, \ldots, M\}$. We also take orthogonal first-layer teacher weights, such that $T$ is the identity matrix. This initialization leads to:

$$R^{km}(0) = 0\,, \qquad \frac{\mathrm{d}}{\mathrm{d}t}W_2^k\bigg|_{t=0} = 0\,, \qquad \frac{\mathrm{d}}{\mathrm{d}t}R^{km}\bigg|_{t=0} = \frac{\sqrt{2}}{\pi\sqrt{1+q_0}}\eta\, f^k(0)\, \tilde{W}_2^m. \quad (27)$$

We can therefore compute the second-layer update to linear order:

$$\frac{\mathrm{d}}{\mathrm{d}t}W_2^k(t) = \frac{2}{\pi^2(1+q_0)}\eta^2\, f^k(0)\, \|\tilde{W}_2\|_2^2\, t + \mathcal{O}(t^2). \quad (28)$$

Eqn. 28 shows that the update of the second-layer weights at early training times is in the direction of the AF matrix, in agreement with the alignment phase observed in experiments. Crucially, it is necessary that $f^k(0) \neq 0$ $\forall k$ in order to have non-zero updates, i.e. the feedback $F$ must not be orthogonal to the first-layer weights at initialization. We now inspect the behavior of the FA at the beginning of training. We have that the update at time zero is:

$$\frac{\mathrm{d}}{\mathrm{d}t}f^k\bigg|_{t=0} = \frac{2}{\pi}\eta\frac{f^k(0)}{\sqrt{(1+q_0)(1+q_f) - f^k(0)^2}}\, (\tilde{W}_2 \cdot \tilde{f}), \quad (29)$$

and

$$f^k(t) = f^k(0) + \frac{2}{\pi}\eta\frac{f^k(0)}{\sqrt{(1+q_0)(1+q_f) - f^k(0)^2}}\, (\tilde{W}_2 \cdot \tilde{f})\, t + \mathcal{O}(t^2). \quad (30)$$

Eqn. 29 illustrates that the feedback-teacher alignment $\tilde{f}$ plays an important role in speeding up the dynamics. Indeed, if $\|\tilde{f}\|_2^2$ is close to zero, the feedback update slows down inducing long plateaus in the generalization error. A similar role is played by the alignment angle between $\tilde{W}_2$ and $\tilde{f}$.

## F PEPITA'S RESULTS COMPARED TO THE BASELINES

Table S1: Test accuracy [%] achieved by BP, FA, DRTP, PEPITA, and PEPITA with WM in the experiments. Mean and standard deviation are computed over 10 independent runs. The nonlinearity is ReLU for all algorithms except DRTP, for which is tanh. WM is used in combination with weight decay with $\lambda = 10^{-4}$ for the networks trained on the CIFAR-10 dataset and for the 3-hidden-layer networks trained on the CIFAR-100 dataset. The other networks are trained without weight decay. Bold fonts refer to the best results exclusively among PEPITA and its improvements.

| | 2 HIDDEN LAYERS | | | 3 HIDDEN LAYERS | | |
|---|---|---|---|---|---|---|
| | MNIST | CIFAR-10 | CIFAR-100 | MNIST | CIFAR-10 | CIFAR-100 |
| BP | 98.85±0.06 | 59.69±0.25 | 32.28±0.17 | 98.89±0.04 | 60.07±0.28 | 32.80±0.16 |
| FA | 98.64±0.05 | 57.76±0.39 | 22.90±0.14 | 97.48±0.06 | 52.99±0.21 | 22.81±0.21 |
| DRTP | 95.36±0.09 | 47.48±0.19 | 20.55±0.30 | 95.74±0.10 | 47.44±0.19 | 21.81±0.24 |
| PEPITA | **98.19**±0.07 | 52.39±0.27 | 24.88±0.15 | 95.07±0.11 | 52.47±0.24 | 01.00±0.00 |
| PEPITA +WD | 98.09±0.07 | 53.09±0.33 | 24.64±0.24 | 95.09±0.16 | 52.56±0.31 | **23.13**±0.21 |
| PEPITA +WM | 98.13±0.05 | **53.44**±0.28 | **26.95**±0.24 | **96.33**±0.12 | **52.80**±0.33 | 23.03±0.28 |

## G SLOWNESS RESULTS

In our analysis of convergence rate, we use the *plateau equation for learning curves* (Dellaferrera et al., 2022):

$$\text{accuracy} = \frac{\text{max\_accuracy} \cdot \text{epochs}}{\text{slowness} + \text{epochs}} \tag{31}$$

By fitting the test curve to this equation, we calculate the slowness parameter, which measures how quickly the network reduces error during training. In mathematical terms, the slowness value corresponds to the number of epochs needed to reach half of the maximum accuracy. In practice, a lower slowness value indicates faster training.

Table S2 shows that the best convergence rate for PEPITA (*i.e.*, small slowness value) is obtained in general by PEPITA with WM (MNIST, CIFAR-100). Compared to the baselines, PEPITA has a better convergence than all the algorithms on MNIST, is the slowest on CIFAR-10, and is the second best after BP on CIFAR-100. However, these results are strongly dependent on the chosen learning rate.

Table S2: Convergence rate in terms of slowness value obtained by BP, FA, DRTP and PEPITA in the experiments for the fully connected models trained on MNIST, CIFAR-10 and CIFAR-100 (same simulations reported in Table 2). PreM refers to pre-mirroring (Sec. 4). The smallest the slowness value, the better the convergence rate. The slowness is computed on the first 60 epochs of the test curve (before the learning rate decay), averaged over 10 independent runs. All the networks are trained without weight decay. The slowness result of FA on CIFAR-10 is lower than in Dellaferrera & Kreiman (2022) as our grid search returned a higher value for the learning rate. Bold fonts refer to the best results exclusively among PEPITA and its improvements.

| | 1×1024 FULLY CONNECTED MODELS | | |
| --- | --- | --- | --- |
| | MNIST | CIFAR-10 | CIFAR-100 |
| BP | 0.061±0.001 | 0.421±0.016 | 1.406±0.053 |
| FA | 0.081±0.002 | 0.463±0.020 | 4.946±0.123 |
| DRTP | 0.059±0.002 | 0.362±0.021 | 12.904±0.443 |
| PEPITA | 0.052±0.004 | 0.894±0.071 | 2.695±0.166 |
| PEPITA+WM | 0.047±0.005 | 0.890±0.081 | 2.333±0.102 |
| PEPITA+WM+PREM | **0.040**±0.002 | **0.856**±0.051 | **1.999**±0.059 |

# H HYPERPARAMETERS

Table S3: 1-hidden-layer network architectures and settings used in the experiments. The nonlinearity is ReLU for all algorithms except DRTP, for which is tanh.

| | 1 HIDDEN LAYER | | | 1 HIDDEN LAYER - NORMALIZATION | | |
| --- | --- | --- | --- | --- | --- | --- |
| | MNIST | CIFAR-10 | CIFAR-100 | MNIST | CIFAR-10 | CIFAR-100 |
| INPUTSIZE | 28×28×1 | 32×32×3 | 32×32×3 | 28×28×1 | 32×32×3 | 32×32×3 |
| HIDDEN UNITS | 1×1024 | 1×1024 | 1×1024 | 1×1024 | 1×1024 | 1×1024 |
| OUTPUT UNITS | 10 | 10 | 100 | 10 | 10 | 100 |
| $\eta$ BP | 0.1 | 0.01 | 0.1 | – | – | – |
| $\eta$ FA | 0.1 | 0.01 | 0.01 | – | – | – |
| $\eta$ DRTP | 0.01 | 0.001 | 0.001 | – | – | – |
| $\eta$ PEPITA | 0.1 | 0.01 | 0.01 | 100 | 10 | 100 |
| $\lambda$ WEIGHT DECAY | $10^{-5}$ | $10^{-4}$ | $10^{-5}$ | 0.0 | 0.0 | 0.0 |
| $\eta$ DECAY RATE | ×0.1 | ×0.1 | ×0.1 | ×0.5 | ×0.1 | ×0.1 |
| DECAY EPOCH | 60,90 | 60,90 | 60,90 | 60,90 | 60,90 | 60,90 |
| BATCH SIZE | 64 | 64 | 64 | 64 | 64 | 64 |
| $\eta$ WM | 0.1 | 0.001 | 0.1 | 0.001 | 0.001 | 0.001 |
| $\lambda$ WEIGHT DECAY WM | 0.0 | 0.1 | 0.5 | 0.1 | 0.1 | 0.1 |
| $\sigma_F^{(0)}$ (UNIFORM) | $0.05{\cdot}2\sqrt{\frac{6}{\text{FANIN}}}$ | $0.05{\cdot}2\sqrt{\frac{6}{\text{FANIN}}}$ | $0.05{\cdot}2\sqrt{\frac{6}{\text{FANIN}}}$ | $0.05{\cdot}2\sqrt{\frac{6}{\text{FANIN}}}$ | $0.05{\cdot}2\sqrt{\frac{6}{\text{FANIN}}}$ | $0.05{\cdot}2\sqrt{\frac{6}{\text{FANIN}}}$ |
| $\sigma_F^{(0)}$ (NORMAL) | $0.05{\cdot}\sqrt{\frac{2}{\text{FANIN}}}$ | $0.05{\cdot}\sqrt{\frac{2}{\text{FANIN}}}$ | $0.05{\cdot}\sqrt{\frac{2}{\text{FANIN}}}$ | $0.05{\cdot}\sqrt{\frac{2}{\text{FANIN}}}$ | $0.05{\cdot}\sqrt{\frac{2}{\text{FANIN}}}$ | $0.05{\cdot}\sqrt{\frac{2}{\text{FANIN}}}$ |
| FAN IN | $28 \cdot 28 \cdot 1$ | $32 \cdot 32 \cdot 3$ | $32 \cdot 32 \cdot 3$ | $28 \cdot 28 \cdot 1$ | $32 \cdot 32 \cdot 3$ | $32 \cdot 32 \cdot 3$ |
| #EPOCHS | 100 | 100 | 100 | 100 | 100 | 100 |
| DROPOUT | 10% | 10% | 10% | 10% | 10% | 10% |

Table S4: 2-, 3-hidden-layer network architectures and settings used in the experiments. The nonlinearity is ReLU for all algorithms except DRTP, for which is tanh. (*) For the 3-hidden-layer network trained with PEPITA on the MNIST dataset, we do not use learning rate decay, as indicated by the grid search.

| | 2 HIDDEN LAYERS | | | 3 HIDDEN LAYERS | | |
|---|---|---|---|---|---|---|
| | MNIST | CIFAR-10 | CIFAR-100 | MNIST | CIFAR-10 | CIFAR-100 |
| INPUTSIZE | $28{\times}28{\times}1$ | $32{\times}32{\times}3$ | $32{\times}32{\times}3$ | $28{\times}28{\times}1$ | $32{\times}32{\times}3$ | $32{\times}32{\times}3$ |
| HIDDEN UNITS | $2{\times}1024$ | $2{\times}1024$ | $2{\times}1024$ | $3{\times}1024$ | $3{\times}1024$ | $3{\times}1024$ |
| OUTPUT UNITS | 10 | 10 | 100 | 10 | 10 | 100 |
| $\eta$ BP | 0.1 | 0.01 | 0.1 | 0.1 | 0.01 | 0.1 |
| $\eta$ FA | 0.1 | 0.01 | 0.01 | 0.01 | 0.001 | 0.01 |
| $\eta$ DRTP | 0.001 | 0.001 | 0.001 | 0.001 | 0.001 | 0.001 |
| $\eta$ PEPITA | 0.1 | 0.01 | 0.01 | 0.001 | 0.01 | 0.01 |
| $\lambda$ WEIGHT DECAY | $10^{-5}$ | $10^{-4}$ | $10^{-5}$ | $10^{-5}$ | $10^{-4}$ | $10^{-4}$ |
| $\eta$ DECAY RATE (*) | $\times 0.1$ | $\times 0.1$ | $\times 0.1$ | $\times 0.1$ | $\times 0.1$ | $\times 0.1$ |
| DECAY EPOCH | 60,90 | 60,90 | 60,90 | 60,90 | 60,90 | 60,90 |
| BATCH SIZE | 64 | 64 | 64 | 64 | 64 | 64 |
| $\eta$ WM | 0.00001 | 1.0 | 1.0 | 0.1 | 0.001 | 0.001 |
| $\lambda$ WD WM | 0.0 | 0.1 | 0.1 | 0.001 | 0.1 | 0.1 |
| $\sigma_F^{(0)}$ (UNIFORM) | $0.05{\cdot}2\sqrt{\frac{6}{\text{FANIN}}}$ | $0.05{\cdot}2\sqrt{\frac{6}{\text{FANIN}}}$ | $0.05{\cdot}2\sqrt{\frac{6}{\text{FANIN}}}$ | $0.05{\cdot}2\sqrt{\frac{6}{\text{FANIN}}}$ | $0.05{\cdot}2\sqrt{\frac{6}{\text{FANIN}}}$ | $0.05{\cdot}2\sqrt{\frac{6}{\text{FANIN}}}$ |
| $\sigma_F^{(0)}$ (NORMAL) | $0.05{\cdot}\sqrt{\frac{2}{\text{FANIN}}}$ | $0.05{\cdot}\sqrt{\frac{2}{\text{FANIN}}}$ | $0.05{\cdot}\sqrt{\frac{2}{\text{FANIN}}}$ | $0.05{\cdot}\sqrt{\frac{2}{\text{FANIN}}}$ | $0.05{\cdot}\sqrt{\frac{2}{\text{FANIN}}}$ | $0.05{\cdot}\sqrt{\frac{2}{\text{FANIN}}}$ |
| FAN IN | $28 \cdot 28 \cdot 1$ | $32 \cdot 32 \cdot 3$ | $32 \cdot 32 \cdot 3$ | $28 \cdot 28 \cdot 1$ | $32 \cdot 32 \cdot 3$ | $32 \cdot 32 \cdot 3$ |
| #EPOCHS | 100 | 100 | 100 | 100 | 100 | 100 |
| DROPOUT | 10% | 10% | 10% | 10% | 10% | 10% |

# I   TRAINING DEEPER FULLY-CONNECTED MODELS

In the field of biologically inspired learning, reaching convergence on networks with more than a couple of hidden layers is a long standing challenge. Two significant reasons for this are the difficulty of learning hierarchical representations and the explosion of weight updates and activities in deep layers Illing et al. (2021). In particular, forward-only learning schemes have been shown to work so far on a maximum of four hidden layers (FF (Hinton, 2022)). The original PEPITA paper was only able to train 1 hidden layer models (Dellaferrera & Kreiman, 2022), therefore our strategy to reach convergence with up to 5 hidden layers (Fig. S5) represents a significant improvement over the previous work.

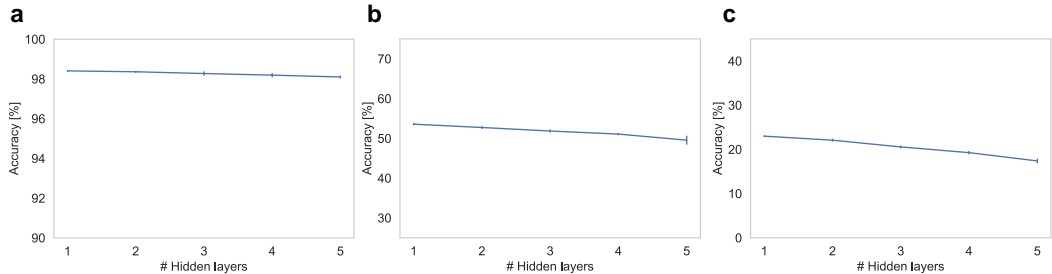

Figure S5: Test accuracy obtained with PEPITA and normalization of the activations for 1- to 5-hidden-layer networks on (a) MNIST, (b) CIFAR-10, and (c) CIFAR-100. Note that compared to Fig. 3 the trend is decreasing, as here we use activation normalization to obtain convergence for 4 and 5 hidden layers, which we did not use for Fig. 3.

## J  WEIGHT MIRRORING

Algorithm S3 describes how WM is applied to PEPITA. This algorithm is applied at the end of each training epoch. $\sigma_F^{(t+1)}$ refers to the standard deviation of the entries of $F(t+1) = \prod_{\ell=1}^{L} F_\ell(t+1)$.

---

**Algorithm S3** Implementation of WM

$\{$Mirror weights$\}$
**for** $\ell = 1, ..., L$ **do**
  $\delta_{\ell-1} \sim \mathcal{N}(\mu, \sigma^2)$
  $\delta_\ell = \sigma_\ell(W_\ell \delta_{\ell-1})$
  $F_\ell(t+1) = F_\ell(t) - \eta \delta_{\ell-1} \delta_\ell^\top$
**end for**
$\{$Normalize feedback matrices$\}$
**for** $\ell = 1, ..., L$ **do**
$$F_\ell(t+1) = \left(\sigma_F^{(0)} / \sigma_F^{(t+1)}\right)^{1/L} \cdot F_\ell$$
**end for**

---

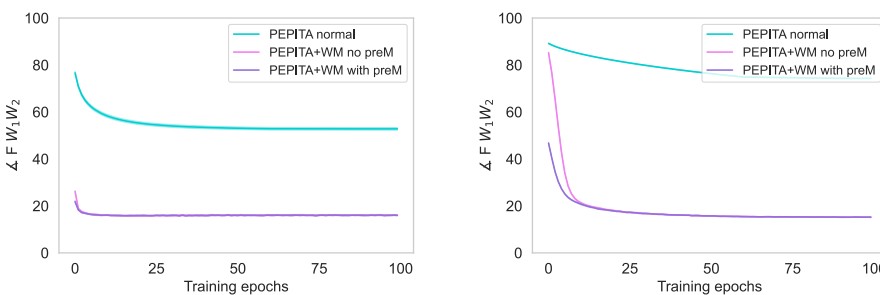

Figure S6: Alignment angle between $F$ and $W_{tot}$ during training with (pink and purple curves) or without (blue curve) WM for the MNIST (*left panel*) and CIFAR-100 (*right panel*) datasets. PreM refers to pre-mirroring (Sec. 4). The hyperparameters are reported in Table S3. The plots indicate mean and standard deviation over 10 independent runs.

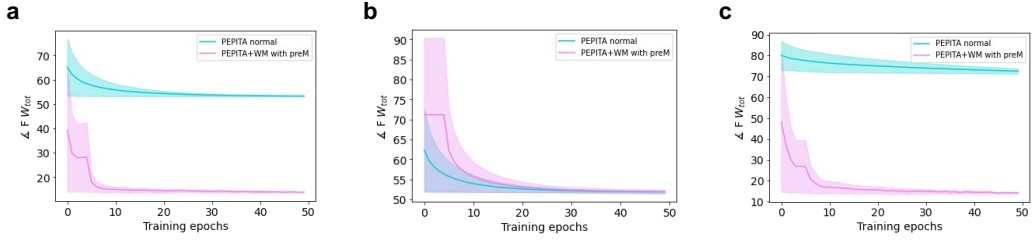

Figure S7: Alignment angle between $F$ and $W_{tot}$ during training with (pink curve) or without (blue curve) WM for the MNIST dataset for (a) 1-, (b) 2-, (c) 3-hidden-layer networks. The hyperparameters are reported in Table S3 for the 1-hidden-layer network and Table S4 for the 2- and 3-hidden-layer networks. The plots indicate mean and standard deviation over 10 independent runs.

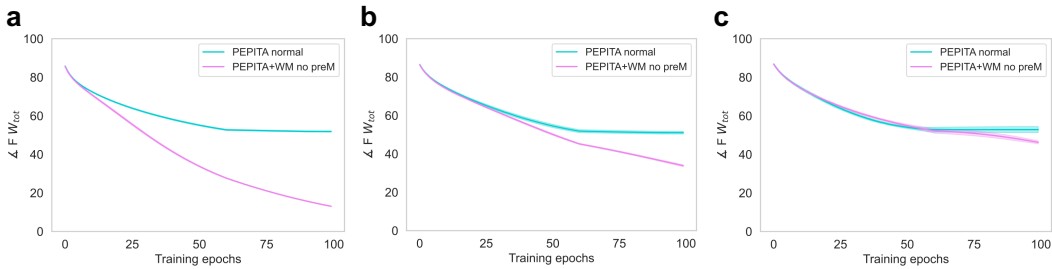

Figure S8: Alignment angle between $F$ and $W_{tot}$ during training with (pink curve) or without (blue curve) WM for the CIFAR-10 dataset for (a) 1-, (b) 2-, (c) 3-hidden-layer networks. The hyperparameters are reported in Table S3 for the 1-hidden-layer network and Table S4 for the 2- and 3-hidden-layer networks. The plots indicate mean and standard deviation over 10 independent runs.

