# OpenReview forum: "Forward Learning with Top-Down Feedback: Empirical and Analytical Characterization"
_ICLR.cc/2024/Conference — ICLR 2024 poster_

### Official Review · Reviewer_knnT · 2023-10-25

**Soundness:** 3 good
**Presentation:** 2 fair
**Contribution:** 3 good
**Rating:** 5
**Confidence:** 3

**Summary:**

This paper provides further analysis of the PEPITA learning rule, a biologically-inspired "forward-only" approach to train neural networks by replacing the backward pass with another forward pass. In connects PEPITA theoretically to two other biologically-inspired approaches, namely an adaptive version of Feedback Alignment and Forward-Forward. Additionally, the paper explores how the initialization of the feedback weights and applying techniques like weight decay, activation normalization, and weight mirroring can improve PEPITA's performance. The experimental findings demonstrate minor improvements through normalization and weight mirroring.

**Strengths:**

- Biologically-inspired and forward-only training approaches are a relevant and timely topic.
- It would be very useful to unify the multitude of different approaches into a common framework. By connecting three major biologically-inspired approaches, this paper takes an important step towards that goal.
- Given the still-remaining gap between backpropagation and forward-only approaches, analyzing and improving their performance is essential for their practical relevance.
- Section 4.2 gives a well-structured summary of the experimental results. It would have been nice, to include a discussion of the impact of weight decay, activation normalization, and weight mirroring

**Weaknesses:**

- The experimental improvement and evaluation of PEPITA on their own offer only limited novelty. However, together with the analysis connecting PEPITA, FA, and Forward-Forward, this paper amounts to a considerable contribution.
- experimental evaluation:
 - only small, fully-connected networks, not that suited to more complex vision datasets
 - only image classification (MNIST, CIFAR10, CIFAR100)
 - no experimental comparison to Forward-Forward
 - only minor improvements in accuracy
- Minor Issues:
 - Figures can be a bit hard to read
  - in Fig 2 (especially 2b) the overlap of PEPITA and AFA is hard to see without zooming in
  - Fig 3 would benefit from more contrasting colors
 - the references to Figure 2c in the last paragraph of Section 3.2 seems to be mismatched with the actual order of subfigures in Figure 2
 - Table 1, line 1 seems to have the wrong citation for FA
 - Spelling:
  - in contribution (vi) "analitically"
  - in the last sentence of Section 1: "gpus" should probably be in uppercase, i.e. "GPUs"

**Questions:**

- Which approach do the authors considers their primary contribution? Table 1 the authors refer to PEPITA-TL as "ours" yet this is only discussed briefly in Section 5.1 and left mostly to future work.
- Why are the combinations "weight-decay + normalization + WM" and "weight-decay + normalization + no WM" missing in Table 2? Can weight decay and activation normalization not be combined?
- In Figure 3c: shouldn't pre-mirroring increase the alignment (thus decreasing the angle) before the first epoch? Yet, in Figure 3c, all three lines seem to start at the same value.

---

> ### Author Response · Authors · 2023-11-18
>
> We thank the reviewer for their insightful comments. The point-to-point answer is provided below.
>
> ---
>
> # Weaknesses
>
> 1. We thank the reviewer for appreciating the value of our theoretical results, beyond simple contributions in performance and accuracy.
>
> Experimental evaluation:
>
> 2. Our work extends previous work that impeded training deeper networks with the original PEPITA method, and so improves on the complexity of networks that PEPITA can work with. Overall, many local or biologically plausible training algorithms in present-day literature are not yet ready to be tested on challenging large-scale datasets and highly complex network architectures.
>
> 3. It is unfortunately common in the literature to use image classification tasks as the initial benchmark of new training algorithms. We share this choice with Hinton’s FF paper and others in the same area (e.g. [Ororbia and Mali, 2023]). We do think it would be interesting to extend this work to other types of tasks, particularly to time-dependent datasets, where a forward-only algorithm can be especially useful for analyzing incoming data in real-time. However, this is best treated in a separate paper, where we can focus on the specific needs of such a case.
>
> 4. At this time, we are not aware of any implementation of Forward-Forward which utilizes feedback connections to generate the negative samples, as proposed by [Hinton, 2022]. Given the significant difference, we decided not to compare PEPITA to Forward Forward with externally generated samples. Despite this, for completeness, we have added the results that FF achieves on MNIST as reported in [Hinton, 2022].
>
> 5. As was pointed out above by reviewer sGVa, the performance of forward-only algorithms has been catching up in recent years, but what is lacking is the theoretical understanding of their learning dynamics. Our aim is not yet to apply PEPITA to real-world tasks but to understand which tools may or may not lead to performance improvements, and why, in order to guide further development. In this respect, small improvements are also informative, and even negative results are valuable lessons.
>
> Minor issues:
>
> 6-7-8. We thank the reviewer for these suggestions, we are currently working to implement them in the paper.
>
> 9. We thank the reviewer for noticing this mismatch. We have now correctly referred to Fig.2d at the end of Sec. 3.2
>
> 10. We thank the reviewer for noticing this. The table now reports Lillicrap 2016 instead.
>
> Spelling:
>
> 11. We thank the reviewer for catching the typo. We have corrected this in the revised version of the manuscript.
>
> 12. We thank the reviewer for the suggestion. In the revised version you will now find “GPUs”.
>
> ---
>
> # Questions
>
> 1. We thank the reviewer for the fair question. We consider our primary contribution the theoretical analysis of the nonintuitive learning dynamics of PEPITA, the study of training techniques that lead to improved performance, and the connections with FF and DFA. Overall, we believe that these different characterizations help us build a common framework to study forward-learning rules. The reason why we included PEPITA-TL in Table 1 is to have a complete comparison. Figure S1 shows that training with PEPITA-TL is possible, but we leave the study necessary to make its performance competitive for future work. Moreover, we have added PEPITA-Hebbian and PEPITA+WM in Table 1.
>
> 2. We thank the reviewer for the question. To find the best hyperparameters for training with activation normalization we run a grid search, which includes searching for the optimal value of weight decay. In all cases with normalization, the grid search returns no weight decay, for this reason, we did not include the combination of normalization and weight decay in the table.
>
> 3. We thank the reviewer for the question. The fact that at the start of the training, the angles look identical is due to the small learning rate for WM that the grid search has returned (Table S3). This causes pre-mirroring to have a small effect on the angle, although the effect is noticeable in terms of slowness (Table S2). For MNIST and CIFAR-100, the grid search returns higher learning rates for WM (Table S3), and the effect is more noticeable both in the angles (Figure S6) and in the slowness (Table S2).

---

### Official Review · Reviewer_uKzr · 2023-10-30

**Soundness:** 2 fair
**Presentation:** 2 fair
**Contribution:** 2 fair
**Rating:** 5
**Confidence:** 4

**Summary:**

This paper analyzes the Present the Error to Perturb the Input To modulate the Activity (PEPITA) learning algorithm, showing its connection to Direct Feedback Alignment (DFA) as well as Forward-Forward (FF) algorithm with negative samples being the “modulated” samples in PEPITA. Some empirical experiments also study PEPITA on deeper networks and with several techniques.

**Strengths:**

This paper studies the connections between several biologically more plausible algorithms, providing insights from empirical and analytical perspectives.

**Weaknesses:**

1. I think this paper requires a clearer definition of the considered “forward-only” algorithms. In the abstract, it is defined as “avoiding a backward pass”. However, PEPITA, different from local FF, indeed requires a direct global error feedback to inputs. If PEPITA is considered as “forward-only”, is DFA also “forward-only”, as it shares the similar form of direct global error feedback without layer-by-layer BP? In the sense of “forward-only” algorithms, I don’t think PEPITA is parallel to FF with local learning.

2. In Table 1, why the considered rule has a much worse performance? It is unclear why we should consider it as it does not introduce additional properties over several algorithms while being worse. Additionally, what’s the detailed definition for “local” and “activity freezing”? For PEPITA, it requires global error feedback to inputs, and storage of activations of the first forward propagation until the second forward propagation ends. Why it is considered “local” and solving “activity freezing”? And why PEPITA outperforms DFA in these aspects?

3. In Section 3.1, the similarity to DFA only holds for the first layer. It is unclear for other layers (Eq. (4)).

4. In Section 4, the networks are still quite shallow and experimental results are quite poor. DFA can scale to large datasets and networks [1], and WM also scales to large-scale scenarios [2].

5. Some derivations are not strict and may be contradictory. For example, in Section 3 and Section 5, it is assumed for several times that $Fe$ is much smaller than $x$ so it can be ignored. In Section 5, however, it is also assumed that the “modulated” samples can be used for negative samples in the FF framework. If the modulation is small enough, why can it formulate a valid negative sample? This largely reduces the reliability of the connection between PEPITA and FF.

Overall, I think there is no enough contribution in empirical or theoretical perspectives.

[1] Direct feedback alignment scales to modern deep learning tasks and architectures. NeurIPS, 2020.

[2] Deep learning without weight transport. NeurIPS, 2019.

**Questions:**

1. In Section 3.2, what’s the difference between the analysis and previous analyses for DFA?

2. In Section 5.1, what’s the connection of the derivation with contrastive Hebbian learning [1] or equilibrium propagation [2], which also takes the form of Hebbian and Anti-Hebbian phases?

[1] Equivalence of backpropagation and contrastive Hebbian learning in a layered network. Neural Computation, 2003.

[2] Equilibrium propagation: bridging the gap between energy-based models and backpropagation. Frontiers in Computational Neuroscience, 2017.

---

> ### Author Response · Authors · 2023-11-18
>
> We thank the reviewer for their insightful comments. The point-to-point answer is provided below.
>
> ---
>
> # Weaknesses
>
> 1. A key difference between DFA and PEPITA/FF is that DFA requires the computation of gradients of activation explicitly. This means that because the feedback is added after the non-linearity, an additional backward computation is needed. In FF and PEPITA, instead, the update only depends on activations computed during the forward pass. We also note that in [Hinton, 2022], the author mentions the possibility of negative samples constructed using top-down connections, which is the case for PEPITA. Moreover, we note that in forward learning algorithms, the error does not travel by itself throughout the network and that the networks continuously respond to inputs (original or modulated/negative), unlike in DFA where a forward pass is followed with a backward pass containing the error info.
>
> 2. We thank the reviewer for the questions. First, we want to clarify the difference between spatial and temporal locality. Spatial locality refers to rules that use only local information (unit activation). In [Nokland, 2016], the author mentions that the update is “almost local”. This is due to the fact that the feedback is added after the nonlinearity, and the update needs to compute the derivative of the activation, causing the update to be non-local (it would be local with linear activation). Instead, PEPITA does not require any computation that does not depend on local signals (unit activation). Temporal locality refers to rules that do not need to store the activations in memory, and are able to compute updates on only temporally-local information. This is the case for PEPITA-TL, which updates the weights on the fly. We note that PEPITA-TL is the only temporally-local rule in Table 1. Second, in DFA - similarly to BP - activity freezing is a consequence of alternating a forward pass (the network responds to an input, and the nodes are “active”) and a backward pass (the error information travels through the network from the top to the bottom layers, but the network does not process any inputs, thus the nodes are “inactive” or “frozen”). In PEPITA and in FF, the error never travels through the network “by itself” to drive the updates. The network always responds to an input in a forward fashion - the input is modulated through the error in PEPITA, the negative samples in FF. Last, we would like to note that this paper is not proposing the PEPITA algorithm, but rather an analytical characterization of the latter. Locality and activity freezing are explained thoroughly in the work by [Dellaferrera & Kreiman, 2022].
>
> 3.  As specified at the beginning of Sec. 3, we focus on two-layer networks for our theoretical results. Sec. 3.1 is aimed at providing an analytical understanding of the alignment phenomenon observed in the paper introducing the PEPITA algorithm [Dellaferrera & Kreiman, 2022] that considered two-layer networks. Our approximation suggests that a perturbative approach could be insightful to understand the PEPITA learning rule in deeper networks. The analytical characterization of the dynamics of deep non-linear networks is to a large extent still an open problem even for networks trained with back-propagation and goes beyond the scope of our work.
>
> 4. We thank the reviewer for pointing out the work of [Launay et al 2020] and [Aktrout et al 2019], which we now cite. The reviewer raises valid points. In general, it has been hard to get better results than using back-propagation in the field. We usually refer to approximations to back-propagation. As one example of many, DFA [Nokland 2016] also started with relatively shallow networks and worse performance than back-propagation and it took a few years for the community to extend it to deeper networks [Launay et al 2020, as cited by the reviewer], and even in these deeper networks, back-propagation generally outperforms DFA (though DFA is a very reasonable approximation, [Laynay et al 2020]). Such approximations can still be interesting due to their biological plausibility, theoretical understanding of learning, and the possibility of implementing them in neuromorphic hardware.

---

> > ### Author Response · Authors · 2023-11-18
> >
> > 5. We respectfully disagree with the reviewer’s point that the magnitude of $Fe$ impacts the connection between PEPITA and FF. First, we point out that the small magnitude of $Fe$ is not an assumption, but it is instead a result that we confirmed empirically. Indeed, the approximation of the PEPITA rule (eq. 6) can only hold if $\|Fe\|\ll\| x\|$. The latter is confirmed by Figure 2a and Figure S3: in this figure, we compare models trained with PEPITA versus identically-initialized models trained with AFA (eq.6). The almost identical training curve confirms that the approximation holds, and consequently that $\|Fe\|\ll\| x\|$. Having proved this, we use the same result (not assumption) for Section 5. The fact that despite the small magnitude the feedback can generate valid negative samples is proven by the empirical results we obtain when training networks (Table 2). To answer “why” this is possible, we present an analysis of the learning mechanism of PEPITA (Section 3), we show that our theory matches the empirical results (Figure 2A-B, Appendix D) and we further elucidate the dynamics of learning, and the alignment mechanism (Appendix E).
> >
> >
> > ---
> >
> > # Questions
> >
> > 1. From a technical point of view, the main difference between the analysis of DFA in [Refinetti et al., 2021a] and our derivation in Sec. 3.2 is that our formalism requires an additional ODE for the dynamics of the adaptive feedback. The emergence of adaptive feedback as a natural order parameter governing the dynamics of learning is a theoretical confirmation of the experimental insight in [Dellaferrera & Kreiman, 2022]. Moreover, our results suggest that a perturbative approach (which is not necessary for DFA) is insightful for the analysis of PEPITA.
> >
> > 2. We thank the reviewer for the interesting question. In both [Xie and Seung, 2003] (equation 2.6) and [Scellier and Bengiom 2017] (Section 2.4) we find very clear similarities with PEPITA-Hebbian in terms of the update rule. Moreover, both algorithms, as well as PEPITA, work on perturbations that are propagated through the network, which treats clean and perturbed signals differently (Hebbian and anti-Hebbian phases). The main difference stems from how the perturbation is propagated through the network: while PEPITA adds the perturbation to the input using a random feedback matrix (thus avoiding the weight transport problem), in both [Xie and Seung, 2003] and [Scellier and Bengiom 2017] the perturbation is fed to the output layer and propagated through weights symmetrical to the forward weights, which is an issue from the biological perspective.

---

> > > ### Comment · Reviewer_uKzr · 2023-11-22
> > >
> > > Thank you for the response. Some of my questions are addressed, e.g., the comparison with DFA and the magnitude of $Fe$, and I raise my score to 5. I have some additional comments.
> > >
> > > 1. In PEPITA, there is also a traveling of error from top to input, while DFA propagates errors to each layer. To what extent the error traveling you think is more acceptable? From the neuroscience perspective, three-factor learning rules with a top-down signal are valid and have been verified [1]. And considering biological spiking neural networks, errors from DFA can be combined with local eligibility traces [2] (the presynaptic activities and derivatives of the nonlinearity of postsynaptic neurons can be maintained in the trace), not necessarily breaking locality. On the other hand, a two stage of “clean” and “modulated” passes poses challenges for neural systems to distinguish the stage for update rules, and while PEPITA-TL is considered temporally local, it requires some temporal information to distinguish the stage for Hebbian or anti-Hebbian, so it is still not fully local.
> > >
> > > 2. If PEPITA-Hebbian mainly differs from previous Hebbian and anti-Hebbian methods considering the weight transport problem, there seems some previous works also consider this problem with random feedback [3].
> > >
> > > [1] Control of synaptic plasticity in deep cortical networks. Nature Reviews Neuroscience, 2018
> > >
> > > [2] A solution to the learning dilemma for recurrent networks of spiking neurons. Nature Communications, 2020.
> > >
> > > [3] Contrastive Hebbian learning with random feedback weights. Neural Networks, 2019.

---

> > > > ### Author Response · Authors · 2023-11-23
> > > >
> > > > We thank the reviewer for the interesting questions and for raising the score. Here are our replies to the additional questions:
> > > >
> > > > 1. We thank the reviewer for the references, which we are now citing. We do not have a clear answer on whether three-stage or two-stage algorithms are more consistent with plasticity in biological systems. Regarding DFA’s locality, we agree that DFA can be considered local in the case of spiking neural networks. The current work and a large number of discussions around approximations to backpropagation focus on non-spiking artificial neural networks. We thank the reviewer for suggesting [1], which we are now citing in our discussion section to explain a possible biological mechanism for PEPITA-TL. Indeed, in [Dellaferrera & Kreiman, 2022], the authors speculate that the projection matrix can be linked to the cortico-thalamo-cortical loops throughout the neocortex, while the first forward pass is computed directly on the input (sensory pathway). Thus, different modulatory signals might be activated during the first and second forward passes. Specifically, the different modulatory signals might indicate whether the activity should be maximized or minimized in the case of PEPITA-TL. The literature supports the anatomical separation between pathways (a summary of which can be seen in Fig. 4 in [Lillicrap et al., 2020]). This view is compatible with the reviewer’s perspective on DFA and with the PEPITA update rule.
> > > > 2. We thank the reviewer for suggesting [3]. We agree with the reviewer about the similarity in Hebbian and anti-Hebbian terms between [3] and PEPITA, although the algorithm in [3] still suffers from the activity-freezing problem and the update-locking problem, similar to FA. Despite these differences, given that the update rules (not considering the feedback path) suggested by the authors in this and the previous comment have a high degree of similarity with PEPITA, we are now citing [Detorakis et al., 2019; Xie & Seung, 2003;  Scellier & Bengio, 2017] in Section 5.1.

---

### Official Review · Reviewer_QDBz · 2023-10-31

**Soundness:** 3 good
**Presentation:** 4 excellent
**Contribution:** 3 good
**Rating:** 8
**Confidence:** 4

**Summary:**

The paper connects several forward pass-only learning methods as biologically plausible alternatives to backprop, analyses their performance and error dynamics, and also proposes Hebbian and temporally local approximations for one of those methods.

**Strengths:**

1. The paper is well-written and does a good job articulating its aims and contributions.
2. The paper provides a theoretical link between several algorithms, and derives error dynamics (for a simple problem) for one of them.
3. The Hebbian/anti-Hebbian approximation works as well as the PEPITA algorithm, which is a good sign for bio plausibility (of both PEPITA and FF, since they work similarly).

**Weaknesses:**

1. Performance on CIFAR10/100 of algorithms is lacking compared to backprop (although surprisingly much less so for CIFAR100). This performance gap would likely increase for larger networks.
2. No experiments with convnets, and I haven't found a justification for that. Since cifar10/100 performance gets much better with introducing convolutions (for backprop), it'd be interesting to look at the performance gap there.

**Questions:**

If I understand correctly, PEPITA-TL differs from PEPITA-Hebbian by one feature: the anti-Hebbian error term is computed for the updated weights. Based on that, I have two questions:
1. Would performance of both algorithms match if we reduce the learning rate of PEPITA-TL? Intuitively it should, since the anti-Hebbian error terms would match.
2. Can we consider PEPITA-Hebbian temporally local since it's a long-term plasticity rule? Since the network is trained with two passes over the same input, the PEPITA-TL approximation adds the first Hebbian term immediately after the first forward pass, so 10s/100s of ms. This would be too short for long-term Hebbian changes (as far as I know), so in your model you can treat both parts of the update rule as if they happen for the same network state -- meaning that PEPITA-Hebbian would be temporally local. (I guess you can account for short-term plasticity as a result of the first forward pass (so non-Hebbian changes proportional to input activity only).)

Small comments:
1. Tab1 should have PEPITA-Hebbian performance too.
2. The appendix can be included in the main pdf in this conference.

---

> ### Author Response · Authors · 2023-11-18
>
> We thank the reviewer for the positive comments on our work. We address the concerns and questions below.
>
> ---
>
> # Weaknesses
>
> 1. Generally, achieving superior outcomes compared to back-propagation in the field has proven challenging. Typically, we resort to utilizing approximations to back-propagation. For instance, DFA [Nokland 2016], among various examples, initially employed relatively shallow networks and exhibited poorer performance than back-propagation. It took a few years for the community to enhance it for deeper networks. We believe that our work provides the insights necessary to improve the performance of these types of algorithms.
>
> 2. In [Dellaferrera & Kreiman, 2022], the authors show that the algorithm can work well on CNNs too. In this work, we focused on other aspects, in particular on the theoretical analysis, which can be performed on fully connected networks. We leave the extension of our results to future work.
>
> ---
>
> # Questions
>
> 1. We have run experiments with PEPITA using the same hyperparameters as the best ones for PEPITA-TL. With this choice of hyperparameters, standard PEPITA also reaches around 40% accuracy, due to the low learning rate. Though these parameters were chosen using a grid search for PEPITA-TL, we conclude that the hyperparameter choice is not sufficient to explain the performance gap.
>
>
> 2. We thank the reviewer for the interesting insight. We provide an answer from two different perspectives. From the hardware implementation perspective, we say that a rule is temporally local if the updates are computed without the need to store any information in memory. This is the case for PEPITA-TL, which computes updates on the fly without any need to keep activations in memory. Instead, PEPITA-Hebbian needs to store the activations of the first forward pass and then use them for the update. From the biological perspective, we believe that the reviewer’s insight is very interesting: on the timescale of neural plasticity, we agree that PEPITA-Hebbian might be temporally local, which would also mean that PEPITA is.
>
> Small comments:
>
> 1. We agree with the reviewer, and we added PEPITA-Hebbian and PEPITA+WM in Table 1.

---

> > ### Comment · Reviewer_QDBz · 2023-11-21
> >
> > Thank you for the response! Having looked at this and other responses/reviews, I keep my current score/recommendation of accept.

---

### Official Review · Reviewer_sGVa · 2023-11-01

**Soundness:** 4 excellent
**Presentation:** 3 good
**Contribution:** 4 excellent
**Rating:** 8
**Confidence:** 3

**Summary:**

Forward-only learning algorithms are an alternative to backpropagation (BP), which are potentially more biologically plausible, more memory efficient, and more computationally efficient.
Two of these forward-only learning algorithms are Forward-Forward (FF) and PEPITA, the Present the Error to Perturb the Input To modulate the Activity learning rule; both algorithms rely on two forward passes to learn, rather than a forward and a backward pass.
However, the learning dynamics of these algorithms are not well understood, and both still include components that are biologically implausible.
In this work, the authors offers an explanation of how PEPITA learns by connecting them to feedback alignment.
They then use their insight to improve the performance of PEPITA by combining it with weight mirroring, a technique to improve performance of feedback alignment, as well as weight decay and activity normalization.
Finally, they propose PEPITA-time-local, an alternative learning rule that improves the biological plausibility of PEPITA by keeping updates local in time (in addition to space), at the cost of some accuracy.

**Strengths:**

- This paper exposes a novel connection between feedback alignment and forward-only algorithms. This connection is particularly interesting because both type of algorithms have improving biological plausibility as a principal goal; exposing underlying relationships may mutually benefit researchers for both types of algorithms. In fact, the authors have already taken a step in this direction, by empirically showing that weight mirroring, a technique developed for feedback alignment, can also be used to improve the performance of PEPITA.
- As the authors also point out, while forward-only learning algorithms have gained traction in recent years, and are gradually catching up to the performance of backpropagation on increasingly complex tasks, theoretical understanding of their learning dynamics is mostly lacking; work that shed light on how forward-only learning works is very welcome.
- The experiments performed are extensive, and the results are convincing.

**Weaknesses:**

- The theoretical analysis focuses only on shallow 2-layer networks, and I’m uncertain that the Adaptive Feedback rule can be easily extended to analyze deeper networks, as the error signal has to pass through multiple hidden layers either forwards (forward-only) or backwards (feedback alignment). This limits the applicability of the analysis, making it disjoint from the experimental results on multi-layer PEPITA networks.

**Questions:**

- Figure 3 shows the performance of PEPITA-Hebbian, PEPITA + weight decay, and PEPITA + weight mirroring on CIFAR-10. Do the same trends hold for MNIST and CIFAR-100?

- I’m a bit confused on how much the PEPITA-Hebbian approximation impacts the accuracy of the model. Figure 3a shows that the impact is very small on CIFAR-10, maintaining over 50% accuracy on CIFAR-10 for 1, 2, and 3 hidden layers, but in Figure S1 the accuracy of a 1-layer PEPITA-TL model drops to around 40%. Could you explain why this is? If this is due to hyperparameter differences, how well would a non-TL PEPITA model compare in performance under the same hypersparameters/conditions as those used for Figure S1?

---

> ### Author Response · Authors · 2023-11-18
>
> We thank the reviewer for highlighting the strengths of our paper. Below, we provide detailed answers to the reviewer's concerns.
>
> ---
>
> # Weaknesses
>
> 1. The reviewer is right in observing that our theoretical analysis is limited to one-hidden layer networks. Our goal is to elucidate the alignment phenomenon initially observed in the paper introducing the PEPITA algorithm [Dellaferrera & Kreiman, 2022], which considers one-hidden layer networks. Further work would be required to study the multi-layer case, but our approximation suggests that a perturbative approach could be useful in this direction. We stress that the analytical frameworks to study the dynamics of learning in deep networks are still very limited, and – to the best of our knowledge – tracking analytically the average performance of deep nets with non-linear activations is still an open problem even for networks trained with back-propagation. One possible preliminary step to elucidate the alignment phenomenon in the multi-layer case would be to consider deep linear networks. We added a comment in Sec. 7.
>
> ---
>
> # Questions
>
> 1. As shown in Table S1, the performance decreases for 3-hidden-layer networks for MNIST (95.07% accuracy) and CIFAR-100 (drops to random chance). For CIFAR-100 weight decay allows us to train the 3-hidden-layer network (23.13% accuracy). We run additional experiments to test PEPITA Hebbian in MNIST and CIFAR-100 with 1-, 2-, and 3-hidden layer networks and we observe less than 0.15% difference (statistically insignificant) in accuracy between PEPITA and PEPITA Hebbian.
>
> 2. In the work by [Dellaferrera and Kreiman, 2022], the authors show that the presynaptic term ($h_{\ell - 1}^{err}$) can be replaced with the same term computer in the first forward pass ($h_{\ell - 1}$) without changing the performance. For this reason, we expected the Hebbian learning rule in eq. 8 to perform equivalently to the original learning rule, which is empirically the case. In the time local version, the updates are computed separately, meaning that the update computed during the second forward pass is operating on the weights that have already been updated in the first forward pass. We suspect that separating the Hebbian and anti-Hebbian updates changes the loss landscape, although we leave this analysis for further work. With the same choice of hyperparameters, standard PEPITA also reaches around 40% accuracy, due to the low learning rate. Though these parameters were chosen using a grid search for PEPITA-TL, we conclude that the hyperparameter choice is not sufficient to explain the performance gap.

---

> > ### Comment · Reviewer_sGVa · 2023-12-04
> >
> > Thank you for the clarifications. I believe this is a strong paper and maintain my stance on acceptance.

---

### Meta-Review · Area_Chair_9zKv · 2023-12-06

**Metareview:**

The paper focuses on forward-only learning algorithms, which are an alternative to backpropagation. The authors draw connections between forward-only algorithms, particularly PEPITA, and feedback alignment. The reviewers generally agree that the insights are useful and the paper is well written. However, they have raised concerns, e.g., about the limited scope of the empirical and theoretical evaluation. The authors have somewhat addressed these concerns, however, as with much of the ‘biologically-plausible’ neural network literature, the results are obtained using small networks on relatively simple datasets and do not completely match/outperform backpropagation.

**Justification For Why Not Higher Score:**

The paper explores a relatively restricted empirical setting. While the results of the paper are interesting, the empirical evaluation is not sufficiently significant to warrant a spotlight or oral.

**Justification For Why Not Lower Score:**

This paper is borderline, with two reviewers recommending acceptance and two reviewers recommending marginal rejection. The main weaknesses pointed out by the reviewers with lower scores pertain to the empirical evaluation. While I share their concerns, I believe that these evaluations are inline with previous works in the literature. Thus, given that the reviewers found the insights of the paper worthwhile, I would tend to favor acceptance.

---

### Decision · Program_Chairs · 2024-01-16

Accept (poster)